# Two-photon absorption under few-photon irradiation for optical nanoprinting

Zi-Xin Liang[1,3], Yuan-Yuan Zhao [1,3] ✉, Jing-Tao Chen[1,3], Xian-Zi Dong[2,3], Feng Jin [2], Mei-Ling Zheng [2] ✉ & Xuan-Ming Duan [1] ✉

Two-photon absorption (TPA) has been widely applied for three-dimensional imaging and nanoprinting; however, the efficiency of TPA imaging and nanoprinting using laser scanning techniques is limited by its trade-off to reach high resolution. Here, we unveil a concept, few-photon irradiated TPA, supported by a spatiotemporal model based on the principle of wave-particle duality of light. This model describes the precise time-dependent mechanism of TPA under ultralow photon irradiance with a single tightly focused femtosecond laser pulse. We demonstrate that a feature size of 26 nm (1/20 λ) and a pattern period of 0.41 λ with a laser wavelength of 517 nm can be achieved by performing digital optical projection nanolithography under few-photon irradiation using the in-situ multiple exposure technique, improving printing efficiency by 5 orders of magnitude. We show deeper insights into the TPA mechanism and encourage the exploration of potential applications for TPA in nanoprinting and nanoimaging.

The two-photon absorption (TPA) process, which involves two quantum transitions originally studied by Maria Göppert-Mayer[1] based on Dirac's dispersion theory[2], has found widespread application in various fields, including nonlinear spectroscopy[3], fluorescence microscopy[4], optical memory[5], lithography[6–8]. In general, the TPA rate of photoactive materials irradiated by a focused laser beam is proportional to the squared light intensity $I^2$. It is typically assumed that the two photons are simultaneously absorbed via a virtual state[9]. Since the absorption cross-section of TPA is low, a tightly focused femtosecond laser beam has been generally used in TPA fluorescence microscopy and lithography.

As a well-established nanoprinting technique, two-photon lithography (TPL) utilizing a femtosecond laser direct writing (LDW) can fabricate arbitrary two-dimensional (2D) and three-dimensional (3D) structures with feature sizes ranging from nanometers to micrometers[10–12]. Leveraging the quadratic nonlinearity of TPA and precise control over processing parameters[13], TPL finds wide-ranging applications in microelectronics[14], optics[15,16], mechanical electric microsystems[17], biomedicine[18–20]. However, with diffraction-limited focusing, the laser peak intensities can reach values as high as $I = 10^{12}$

W cm$^{-2}$, accompanied by corresponding photon irradiance of $3 \times 10^{33}$ s$^{-1}$ cm$^{-2}$ that is sufficient to enable appreciably effective TPA[21,22]. Such high photon irradiance can easily trigger high-order nonlinear optical processes, leading to photobleaching, micro-explosions and a narrowed process window[23]. Furthermore, TPA only occurs in the tiny area of the focused laser spot, resulting in low throughput. Although multi-focus[24–27], techniques can partially improve fabrication efficiency, the serial point-by-point writing protocol of LDW remains inadequate for efficiently fabricating structures with multiscale components, ranging from nanoscale to macroscale.

To improve the manufacturing efficiency of TPL, two-photon digital optical projection nanolithography (TPDOPL) technology has been developed[28]. This method utilizes a digital micromirror device (DMD) as a digital mask[29] which can be easily changed by replacing the data of the digital mask with millions of pixels. The throughput of TPDOPL significantly exceeds that of LDW by several orders of magnitude[29–31]. Meanwhile, a resolution of 32 nm, equivalent to 1/12 of the laser wavelength, was achieved by inducing TPA under irradiation with a laser peak intensity of only $1.40 \times 10^5$ W/cm². This corresponds

[1]Guangdong Provincial Key Laboratory of Optical Fiber Sensing and Communications, Institute of Photonics Technology, Jinan University, Guangzhou, China. [2]Laboratory of Organic NanoPhotonics and CAS Key Laboratory of Bio-Inspired Materials and Interfacial Science, Technical Institute of Physics and Chemistry, Chinese Academy of Sciences, Beijing, China. [3]These authors contributed equally: Zi-Xin Liang, Yuan-Yuan Zhao, Jing-Tao Chen, Xian-Zi Dong. ✉e-mail: yyzhao@jnu.edu.cn; zhengmeiling@mail.ipc.ac.cn; xmduan@jnu.edu.cn

to a photon irradiance of $2.8 \times 10^{23}$ s$^{-1}$ cm$^{-2}$, which is almost 10 orders of magnitude lower than that required for LDW[32,33]. Notably, the number of irradiated photons per pulse per pixel ($N_{spp}$) for a single DMD pixel was even as low as 5, demonstrating that TPA can be effectively triggered under conditions of ultralow-photon irradiance, which we define here as few-photon irradiation.

Here, we introduce a concept, few-photon irradiated TPA (*fp*TPA), offering a perspective on the TPA process and its probability distribution under ultralow photon irradiance with a tightly focused femtosecond laser pulse. We developed a spatiotemporal model based on the principles including wave-particle duality, uncertainty and photon distribution in a femtosecond laser pulse to describe the precise time-dependent mechanism of TPA. Through simulations using this model, we determined the probability and distribution of effective TPA (*e*TPA) as a function of varying photon irradiance. To validate the *fp*TPA concept and spatiotemporal model, we conducted TPDOPL experiments, which achieved a feature size of 26 nm, equivalent to 1/20 of the wavelength ($\lambda$), and improved patterning efficiency by 5 orders of magnitude, effectively breaking the trade-off shackle between resolution and efficiency in TPL. Additionally, we proposed and developed the in-situ digital multiple exposures (*i*DME) method, enabling fine, dense, and complex patterning with TPDOPL. Here, we show the underlying physics of the *fp*TPA and demonstrate TPDOPL as a versatile and powerful tool for fabricating devices with high resolution, efficiency and accuracy in the fields of microelectronic integrated circuits, optical waveguides, and biological microfluidics.

## Results

### Few-photon irradiated two-photon absorption

TPA is widely recognized as the simultaneous absorption of two photons via a virtual state[1]. From the perspective of quantum electrodynamics[31], the process begins with the absorption of a photon, transitioning the photosensitive molecular system from the ground state (**g**) to an intermediate state (**i**). The absorption occurring at the intermediate stage without energy conservation is referred to virtual absorption, and the intermediate state is known as the virtual state. Subsequently, a second photon is absorbed, completing the transition to the final excited state (**e**). The total energy is conserved, resulting in $E_{TPA} \approx 2h\nu$. To illustrate this, we use a simplified photon diagram, where photons with energy $h\nu$ and polarization $k$ are depicted as point-like particles. The interaction between the photon (with energy $h\nu$) and the molecule can be represented by the time-ordered graph for TPA, as shown in the inset of Fig. 1a (detail shown in Supplementary Note 1).

The virtual state does exist; however, it does not remain populated long enough for the second photon to interact with the molecule before the virtual state "decays". Therefore, the molecule can absorb two photons arriving at different times simultaneously when the time interval $t_0$ between the arrival of the two photons is less than the virtual state lifetime, $\tau_i$, as illustrated in Fig. 1a. An estimation of the intrinsic lifetime of the virtual state, $\tau_i$, can be obtained using Heisenberg's uncertainty principle and the single intermediate state approximation[34],

$$\tau_i = h\left(4\pi\Delta\widetilde{\nu}_{ie}\right)^{-1} \tag{1}$$

where $h$ is Planck's constant and $\Delta\widetilde{\nu}_{ie}$ is the energy difference between the photon energy of $h\nu$ and the stationary energy of the lowest excited state of the molecule ($E_1$). The values of $\tau_i$, calculated using Eq. (1) with the absorption cut-off wavelength of the photoresist AR-N-7520 utilized in this study, were confirmed to be 0.8 fs and 0.3 fs for incident laser wavelengths of 400 nm and 517 nm, respectively, as depicted in Fig. 1b. The relationship between $\tau_i$ and $\Delta\widetilde{\nu}_{ie}$ is illustrated in

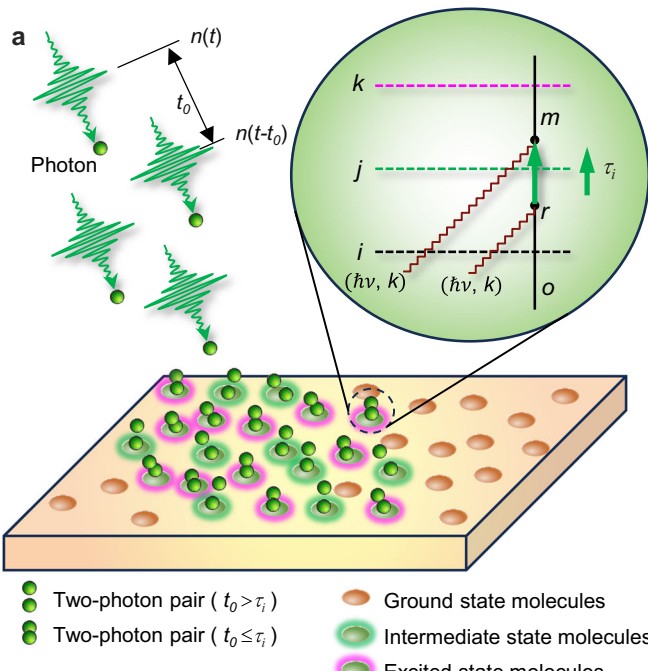

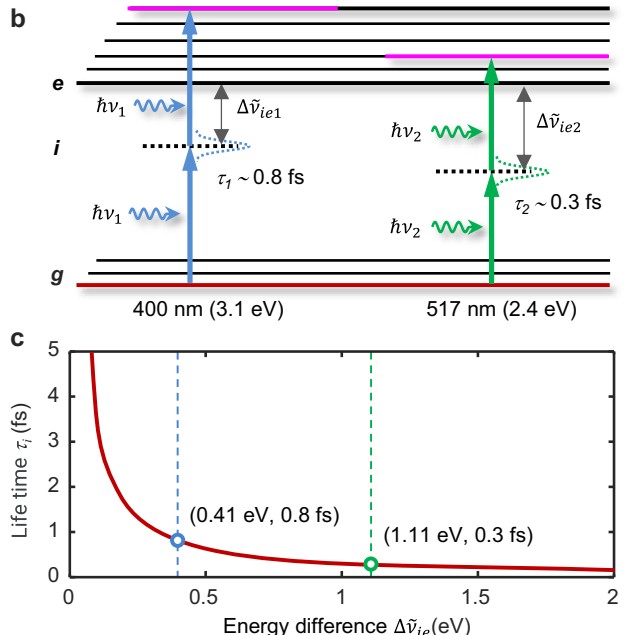

**Fig. 1 | Time-dependent quantum mechanism of two-photon absorption. a** Time-ordered graph for the absorption of two photons of the same mode, in which the graph time flows upwards, the vertical line represents the change taking place in the molecule during the process, and the wave lines represent the photons with the mode ($h\nu$, $k$) ($t_0$ represents the time interval between the arrival of two photons, $\tau_i$ denotes the lifetime of virtual state, $i,j,k$ stands for different moments, $o,r,m$ represents different moments within the molecule). **b** Schematic representations of the energy-level diagram of the two-photon absorption process exciting an electron from the ground state to an excited state passed through an intermediate virtual state under the irradiation of photons with wavelengths of 400 nm and 517 nm, respectively ($\Delta\widetilde{\nu}_{ie}$ represents the energy difference, $g$ denotes the initial state, $i$ denotes the intermediate state, and $e$ denotes the final excited state). **c** The relationship between intrinsic lifetime of the virtual state ($\tau_i$), and the energy difference between the photon energy and the stationary energy of the appropriate low-lying allowed singlet state according to Eq. (1).

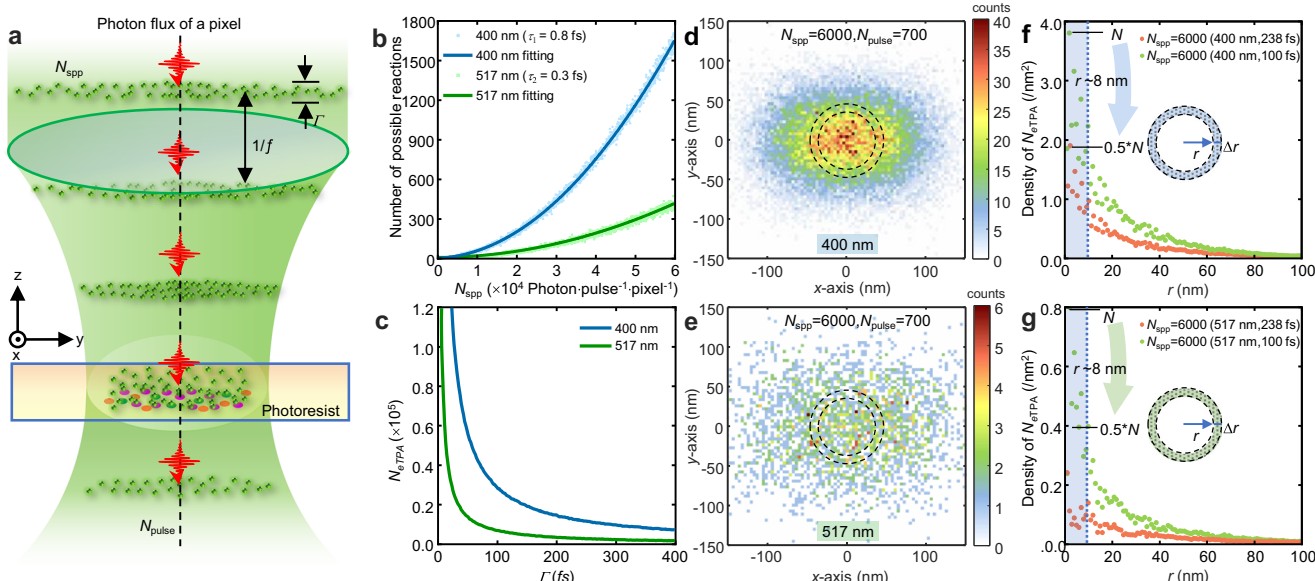

**Fig. 2 | Spatial distribution of $fp$TPA. a** Schematic representations of two-photon absorption of molecules irradiated by the photon flux of a pixel light field from femtosecond pulses ($f$ denotes the repetition rate, and $\Gamma$ represents the pulse width). **b** Relationship between both numbers of triggerable $e$TPA ($N_{e\text{TPA}}$) and inputted photons ($N_{\text{pulse}}$) for a single pulse. **c** Relationship between $N_{e\text{TPA}}$ and pulse width of laser ($\Gamma$). Calculated distributions of $e$TPA on the focused laser spots using the wavelengths of 400 nm (**d**) and 517 nm (**e**) after 700 pulses irradiated with $N_{\text{spp}} = 6000$. The calculated distribution of $e$TPA density in the ring located at a distance '$r$' from the center within the focused spot with wavelengths of 400 nm (**f**) and 517 nm (**g**) ($N_{\text{spp}} = 6000$, $N_{\text{pulse}} = 700$). The blue-shaded region indicates the spatial region with a radius of $r = 8$ nm.

Fig. 1c, clearly demonstrating that the values of $\tau_i$ are significantly dependent on $\Delta\tilde{\nu}_{ie}$.

The probability of TPA is critically influenced by the spatial and temporal distribution of incident photons from a tightly focused femtosecond laser pulse. Each photon can be treated as a discrete event, with its position adhering to a Poisson distribution within the focal spot. Figure 2a illustrates a model depicting the interaction between a tightly focused photon stream, characterized by a repetition frequency $f$, pulse width $\Gamma$, and photosensitive molecules within the photoresist. Considering the time-dependent mechanism of TPA, we hypothesize that two photons, each with energy $\hbar\nu$ where $\hbar\nu < E_1 \leq 2\hbar\nu$, continuously interact with the same molecule within the time interval $\tau_i$ (Fig. 1a). This interaction facilitates the effective TPA ($e$TPA) of the molecule. The time-averaged number of $e$TPA at position $r$ and time $t$ is given by

$$\langle n_{TPA}(r,t) \rangle = \left(\frac{\lambda}{hc}\right)^2 \delta_{g,i}\delta_{i,e}\langle I(r,t) \cdot I(r,t-t_0) \rangle \tag{2}$$

where $\delta_{g,i}$ and $\delta_{i,e}$ are the single-photon absorption coefficients from $g$ to $i$ states and $i$ to $e$ states, respectively. $I(r,t)$ and $I(r,t-t_0)$ represent the energy flow density separated by time $t_0$ at position $r$. $I(r,t)/(hc/\lambda) = n(r,t)$ is defined as the photon flow density. The precise spatial position of a photon as an individual particle colliding on the focal plane is uncertain. However, the distribution of incident photons can be represented by the light intensity distribution of a tightly focused laser spot, as calculated by the point spread function (PSF) (for details on the light intensity distribution, see Supplementary Note 2 and Supplementary Fig. 1). Similarly, the exact timing of an individual photon reaching the focal plane within the pulse duration $\Gamma$ is also uncertain, though the hyperbolic secant function (HSF) can describe the temporal profile of the femtosecond laser pulse. Consequently, the potential distribution of $e$TPA under few-photon irradiation with a femtosecond laser pulse must incorporate both spatial and temporal uncertainties associated with the photons in the pulse.

We employ the Monte Carlo method to simulate the spatial and temporal stochastic processes of photons within a femtosecond pulse, coupled to a focusing system with a numerical aperture (NA) of 1.49 (for detailed quantum model, see Supplementary Note 3 and Supplementary Fig. 2). Each focused photon beam originates from a pixel in the graphics generator, such as DMD, and the sampling area on the focal plane is defined as 3 nm × 3 nm in the simulation. Subsequently, the number of $e$TPA ($N_{e\text{TPA}}$) is calculated by integrating the spatial and temporal methods as described above (see Supplementary Note 4-7 for details).

The triggerable $N_{e\text{TPA}}$ increases quadratically with the $N_{\text{spp}}$ at wavelengths of 400 nm and 517 nm (Fig. 2b), consistent with the $I^2$ relationship. The shorter pulse width of the femtosecond laser enhances $N_{e\text{TPA}}$ (Fig. 2c) by temporally increasing the probability of effective second photon absorption. Our simulations indicate that achieving reliable $e$TPA a single pulse requires approximately 1000 and 1800 photons ($N_{\text{spp}}$) at wavelengths of 400 nm and 517 nm, respectively, with a pulse width of 238 fs. When the pulse width narrows to 100 fs, the required $N_{\text{spp}}$ decreases to 400 and 1000, aligning with the temporal distribution of photons in an ultrashort pulse laser. The $N_{e\text{TPA}}$ at wavelengths of 400 nm is nearly seven times greater than at that of 517 nm when using the same pulse width (refer to Supplementary Fig. 3 and Supplementary Table 2). Reducing the pulse width from 238 fs to 100 fs results in a 2.4-fold increase in $N_{e\text{TPA}}$. This suggests that shorter pulse widths significantly enhance the probability of triggering $e$TPA. The efficiency of photon conversion to $N_{e\text{TPA}}$ depends on $\tau_i$; longer $\tau_i$ values yield higher $N_{e\text{TPA}}$. This correlation with $\Delta\tilde{\nu}_{ie}$ (Fig. 1c) implies that incident photon energy closer to $E_{\text{lowest}}$ likely increases $N_{e\text{TPA}}$ due to extended $\tau_i$.

The spatial and temporal stochasticity of photons at the focal spot determines the potential distribution of $N_{e\text{TPA}}$. Under few-photon irradiation, the randomness in the distribution of $e$TPA decreases as $N_{\text{spp}}$ increases, as illustrated in Supplementary Fig. 4. A lower $N_{\text{spp}}$ results in a higher variance coefficient for the probability of $e$TPA occurrence (Supplementary Table 1), attributable to quantum random noise (Supplementary Fig. 4). Nonetheless, while pulse accumulation

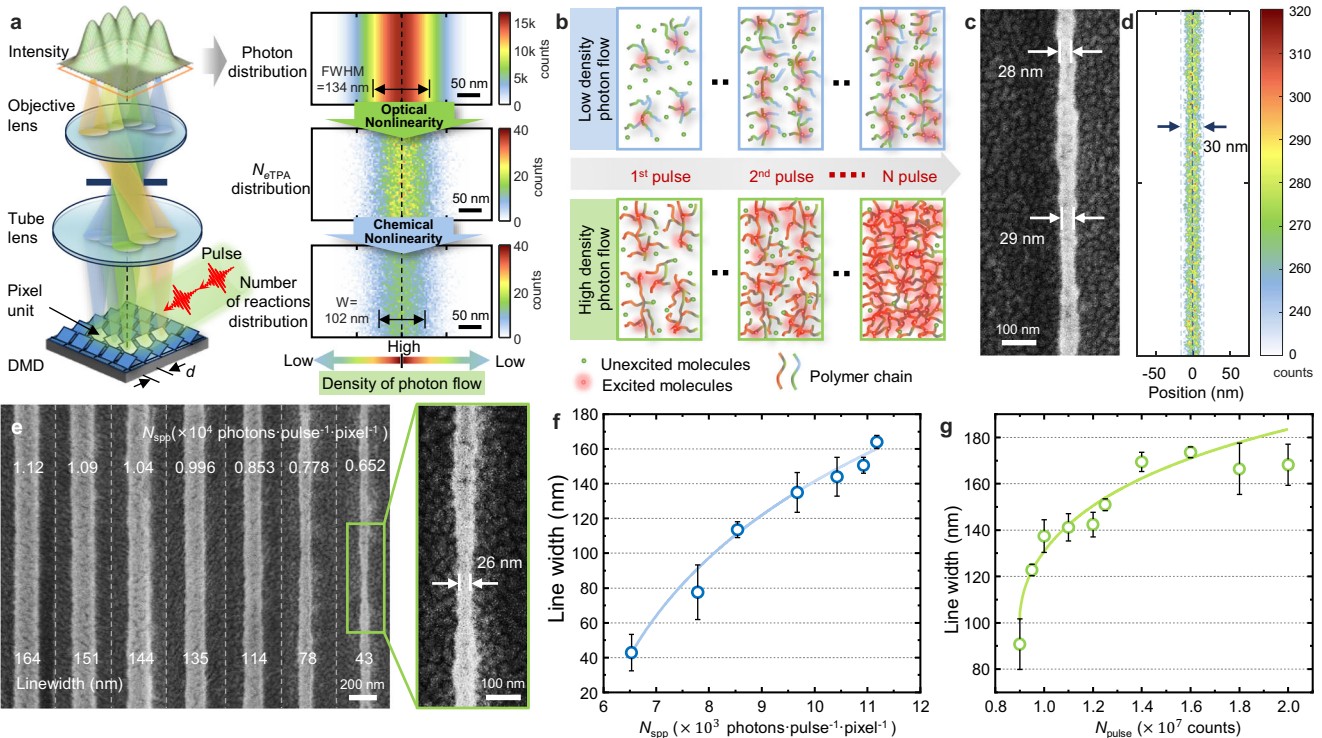

**Fig. 3 | Line width narrowing mechanism for achieving nanoscale resolution.**
**a** Schematic of the TPDOPL system with few-photon irradiation including simulation of photon distribution, $N_{eTPA}$ distribution and reaction site number of the photosensitive molecule distribution under low-photon irradiation ($N_{spp} = 1.25 \times 10^3$, $N_{pulse} = 1 \times 10^4$). **b** Schematic for the narrowing mechanism of the stepwise photopolymerization under femtosecond pulse irradiation. The polymerization degree is modulated by pulse number and photon density. **c** SEM image of the polymer line irradiated by $N_{spp} = 5.14 \times 10^3$ (1.97 fJ/(pulse·pixel)) with $N_{pulse} = 8.5 \times 10^7$. **d** The simulated distributions of $eTPA$ with $N_{spp} = 5.14 \times 10^3$ and $N_{pulse} = 8.5 \times 10^3$. **e** SEM

images of the polymer lines irradiated under different $N_{spp}$, with $N_{pulse} = 6 \times 10^7$. The magnified SEM image is the polymer line irradiated with $N_{spp} = 6.52 \times 10^3$ (2.51 fJ/ (pulse·pixel)) and $N_{pulse} = 6 \times 10^7$. The smallest feature size is 26 nm and the average line width is 43 nm with a standard deviation of 7 nm and roughness of 3-4 nm. **f** The relationship of the line width with a single pixel array as a function of photon flux density. **g** The relationship of the line width exposure with a single pixel array of irradiated pulse numbers. Error bars represent mean ± SD based on 10 independent measurements for each data point.

increases $N_{eTPA}$, it does not affect the efficiency of triggerable $eTPA$ for each pulse (Supplementary Fig. 5). Next, we focus on the spatial distribution of $eTPA$. Typical examples with an $N_{spp}$ of 6000 and an irradiated pulse number ($N_{pluse}$) of 700 are shown in Figs. 2d and 2e (and Supplementary Fig. 6) for wavelengths of 400 nm and 517 nm, respectively. The statistical densities of $N_{eTPA}$ ($d_{eTPA}$) are presented in Figs. 2f and 2g. We calculated the $N_{eTPA}$ within a circular belt of 4 nm width and divided it by the area of the belt. The $d_{eTPA}$ sharply decreases from the center of the focal spot, reaching approximately half its value at a radius of 8 nm, independent of wavelength (Fig. 2f-g). This result indicates that the resolution of TPA under few-photon irradiation can significantly surpass the resolution of the employed wavelength.

**Two-Photon Optical Projection Nanolithography**
To evaluate the effectiveness of our proposal concept and spatiotemporal model, we conducted TPDOPL using a femtosecond pulse laser and a DMD (Fig. 3a, Supplementary Fig. 7). The DMD, with a megapixel-resolution projection layout of arbitrary features, is irradiated by a flat-top beam and focused onto a photoresist film on a cover glass using an oil-immersion objective lens (Nikon, 100×, NA 1.49). The optimal placement angle was determined based on insights gained from our previous studies[35] on the diffraction efficiency of the DMD (Supplementary Fig. 7b). We chose a commercially available non-chemically amplified (non-CA) negative photoresist (AR-N 7520) (phenolic resin acts as the matrix, with bisazide compounds serving as cross-linking agent) because its degree of polymerization, driven by the stepwise photopolymerization mechanism, is easily controllable

and quantifiable for $N_{eTPA}$ under few-photon irradiation. This resist has an absorption peak at 323 nm and an absorption cut-off wavelength of 353 nm (Supplementary Fig. 8), ensuring that only TPA occurs when using femtosecond pulse lasers at both 400 nm and 517 nm wavelengths. Note that the number of excitable molecules in the photoresist by TPA should be less than the calculated $N_{eTPA}$ from the proposed model. The calculated $N_{eTPA}$ predicts the possible opportunity and distribution of $eTPA$ under photon irradiation from the viewpoint of incident photons, but it ignores the molecular concentration, distribution, and quantum yield of TPA in the photoresist. Furthermore, the conversion efficiency from excited molecules by TPA to the practically initiated coupling reaction between molecules should be considered. The number of reaction sites of the photosensitive molecules is ultimately limited by the final absorbed $N_{eTPA}$ and their quantum yield to initiate the reaction.

We utilized incident light with an average power of 1 mW after passing through the objective lens as an illustrative example, specifically with parameters $N_{spp} = 1250$ (0.48 fJ/(pulse·pixel)) and $N_{pulse} = 1 \times 10^7$. The distribution of $N_{eTPA}$ for a line composed of one pixel demonstrates a notable 24% reduction in full width at half maximum (FWHM) compared to the photon distribution (Fig. 3a). According to photopolymerization theory[36,37], the relationship between the concentration of photosensitive molecules (M) excited by TPA and the photon distribution is nonlinear. As the reaction step is repeated, the molecular weight at the center of the exposure field increases exponentially due to the chemical cross-linking reaction. The crosslinking degree of the monomers is controlled by the exposure dose[38,39]. The concentration

distribution of photosensitive molecules involved in the reaction is illustrated in Fig. 3b (Supplementary Note 3 for more details). The exponential increase in molecular weight leads to faster gelation at the exposure field's center, forming insoluble polymer networks more quickly than in the surroundings. Consequently, the superposition and coordination of optical and chemical nonlinearity can effectively reduce the feature size in TPDOPL under few-photon irradiation.

To investigate the effectiveness of the proposed spatiotemporal model for TPDOPL under few-photon irradiation, we fabricated separate lines using our TPDOPL system with a femtosecond laser wavelength ($\lambda$) of 517 nm and pulse width of 238 fs. Using a single-pixel DMD layout, a line with an average width of 41 nm and a minimum feature size of 28 nm (Fig. 3c) was achieved under the irradiation of a total incident photon number per pixel of $4.37 \times 10^{11}$ (0.167 μJ) with accumulation $N_{pulse}$ of $8.5 \times 10^7$ pulses containing $N_{spp}$ of $5.14 \times 10^3$ (1.97 fJ/(pulse·pixel)). Correspondingly, we calculated the $e$TPA distribution using the same photon flux as the experimental result in Fig. 3c but only performed $8.5 \times 10^3$ pulses in simulation. The simulation result shown in Fig. 3d indicates that the $e$TPA distribution is concentrated in a central area of about 30 nm. This validates the effectiveness of our spatiotemporal model for predicting the feature size of TPDOPL under few-photon irradiation.

Photon irradiance density and the accumulated pulse numbers critically influence the line width of TPDOPL. By decreasing $N_{spp}$ from $1.12 \times 10^4$ (4.30 fJ/(pulse·pixel)) to $6.52 \times 10^3$ (2.51 fJ/(pulse·pixel)), the average line width of the polymer line was reduced from 164 nm to 43 nm under the accumulation of $6 \times 10^7$ pulses, achieving a minimum feature size of 26 nm (1/20 $\lambda$), as shown in Fig. 3e. The $N_{eTPA}$ under different $N_{spp}$ irradiations can be observed in Supplementary Fig. 10. The relationship between the polymer line width and photon irradiance density is depicted in Fig. 3f, indicating that the feature size can be reduced by decreasing the photon irradiance density. However, lower photon irradiance density may increase line roughness due to quantum noise, which can increase edge roughness for fine lines (Supplementary Fig. 11). On the other hand, increasing the accumulation of pulses with a fixed photon flux density leads to a widening of the line width, as shown in Fig. 3g.

Another significant aspect pertains to periodic lines in photolithography, which determine the potential feature density achievable in device applications. Our setup corresponds to "the general Sparrow criterion" for parallel projection lithography, where $d = \lambda/NA$. Further detailed in Supplementary Note 13 and Supplementary Fig. 12. Generally, the minimum distinguishable period between adjacent lines is dependent on the wavelength and determined by the equation $HP$ (half pitch) $= 0.5\,\lambda/NA$, following the Sparrow criterion[40]. When the design pattern period is less than the minimum resolvable distance between two lines, double patterning lithography (DPL) can overcome this problem[41]. For instance, at $\lambda = 517$ nm and NA = 1.45, the criterion yields an approximate value $HP_{limit}$ of 174 nm. We designed a line array using the DMD pixel period of 7.56 μm combined with 2 pixels on and 1 pixel off periodically (Supplementary Fig. 13a), corresponding to a period of 226.8 nm. Using irradiation conditions with $N_{pulse} = 6 \times 10^7$ and $N_{spp} = 1.52 \times 10^4$ (5.84 fJ/(pulse·pixel)), the lines were indistinguishable (Supplementary Fig. 13. c). We efficiently utilized the flexibility of TPDOPL by using a DMD as a digital mask, enabling in-situ digital multiple exposures (iDME) to print dense features without being constrained by the diffraction limit. It's a digital dual-exposure technique that eliminates the need for alignment to enhance lithographic resolution in TPDOPL. Using computer-controlled DMD to generate low spatial frequency, sparse 'digital mask' patterns and alternating dual exposures, we double the density of nanopatterns. Since the spacing between DMD micromirrors is fixed, alignment errors are eliminated, making multiple exposures on a single photoresist coating feasible without physical mask alignment steps − surpassing the diffraction limit achievable with single exposures in traditional lithography. Exploiting DMD characteristics, two split layouts with a period of 2'p' are sequentially loaded in situ for double exposure, achieving an exposure result with a period of 'p', as depicted in Fig. 4a. Under twice alternating exposure of $N_{pulse} = 6 \times 10^7$ and $N_{spp} = 8.53 \times 10^3$ (3.28 fJ/(pulse·pixel)), we successfully achieved a dense line array with a period of 210 nm (0.41$\lambda$, $HP = 105$ nm $\approx 0.3\,\lambda/NA < HP_{limit}$), a linewidth of 150 nm, and a gap spacing of 60 nm, as shown in Fig. 4b, surpassing the diffraction limit. Further detailed in Supplementary Note 13 and Supplementary Fig. 12.

Taking advantage of TPDOPL-iDME, we can achieve distinguishable dense structure patterning. When the pitch is less than 5 pixels, a single exposure cannot meet the resolution consistent with the design pattern (Supplementary Fig. 14). Supplementary Fig. 9 illustrates the system's capability for high-throughput operation, enabling uniform exposure of $80 \times 100$ μm$^2$ in a single exposure field and achieving a fabrication rate of $1 \times 10^{-3}$ mm$^2$/s. Figure 4c shows a typical circuit layout selected from a commercial chip design, including isolated and dense lines with a width of 3 or 7 pixels and intervals of 1 and 2 pixels between lines (Supplementary Fig. 15). We employ algorithms[42] to strategically distribute polygons with interspacing distances below 2 pixels across distinct sub-masks, optimizing their arrangement for TPDOPL-iDME. SEM images show that direct single exposure causes indistinguishable results in dense line areas (Fig. 4d). By splitting this layout into two (Fig. 4e) and performing our TPDOPL-iDME approach, we successfully achieved the expected circuit patterning (Fig. 4f). The dense lines are clearly distinguished, and the periods agree well with the design. Furthermore, by optimizing exposure parameters and layout design for TPDOPL-iDME, line width, period, and gap distance can be controlled for finer and denser feature patterning.

Optical devices with curved and circular microstructures have been fabricated using TPDOPL, such as patterns including arrayed waveguide gratings and micro-ring resonators[43]. The radius of the ring affects the value of the free spectral range, and the gap or spacing between the guide and the ring affects the coupling ratio between the waveguide and the ring[44]. Through layout design and the TPDOPL-iDME method, we can fabricate micro-ring filters with varied radius pitches. The widths of the circular rings can be adjusted from 220 nm to 346 nm by increasing $N_{pulse}$ under the irradiation of $N_{spp} = 8.53 \times 10^3$ (3.28 fJ/(pulse·pixel)) (Supplementary Fig. 16a). We patterned the line waveguides, followed by the fabrication of circular rings with different diameters (Fig. 5a), leveraging TPDOPL-iDME. The gap distances between the line and circular rings can be finely adjusted from 66 nm to 480 nm (Supplementary Fig. 16b), optimizing the structures and improving the properties of photonic resonance devices.

The flexibility of TPDOPL-iDME allows us to create arbitrary patterns with various sizes, shapes, and densities, applicable not only in microelectronics and microphotonics but also in microfluidics[45,46]. Microfluidics in microbiology offers an in vitro platform for interactions among diverse cell types, enabling real-time observation and assessment of reaction processes[47]. We designed a rectangular module to substitute the cell chamber and a circular module to replace the cell secretion chamber, with channels of varied sizes to facilitate the addition and observation of multiple cell types and their reactions[48]. Figure 5b shows complex patterns of biological microfluidics fabricated by TPDOPL-iDME, with a total layout size of $120 \times 60$ μm$^2$. The design includes square cell incubators ($3 \times 3$ μm$^2$), rectangular cell chambers ($2.8 \times 6$ μm$^2$), and circular cell collectors with micrometer and sub-micrometer scales are connected by different channels with widths from 70 nm to 800 nm (Fig. 5b iii), effectively carrying and separating viruses of different sizes. Most biomolecular analytes are below microns in size[49], especially foreign objects such as viruses[50], which are usually 20-300 nm in size. Cross-scale biological microfluidics, from micrometer to nanometer, hold promise for providing research platforms for diagnostic and therapeutic methods for viruses like the coronavirus.

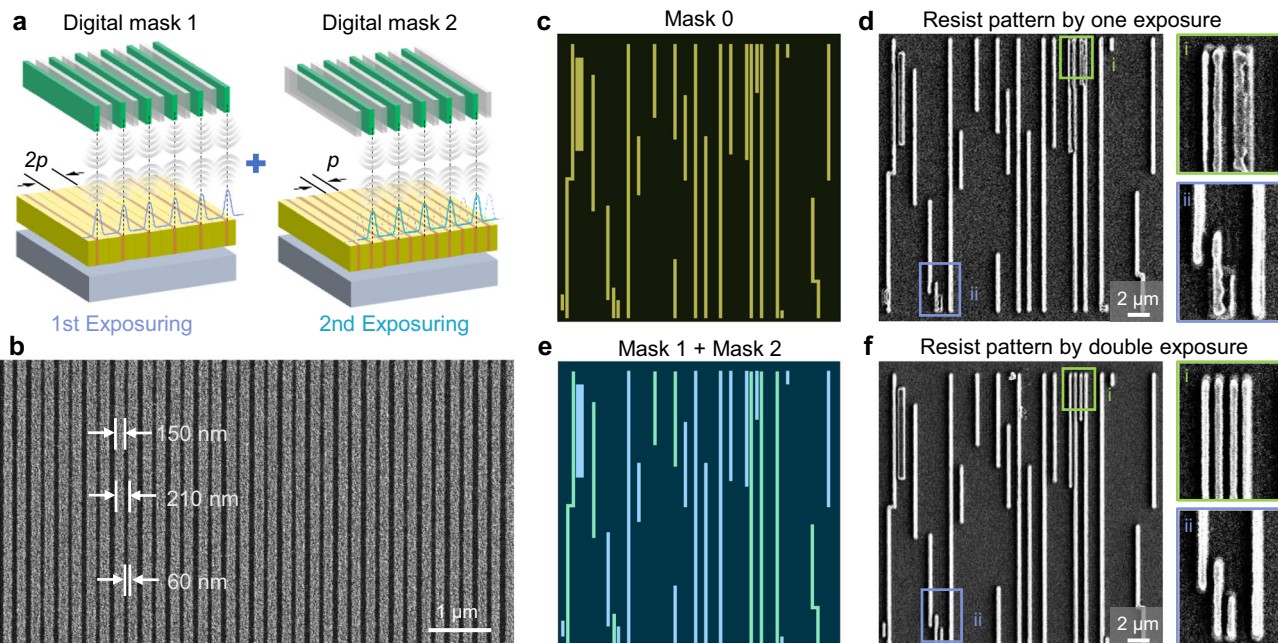

**Fig. 4 | In-situ Digital multiple exposures (*i*DME) and the applications for dense structure patterning. a** Schematic diagram of in-situ digital double exposure. **b** SEM image of dense line patterns with a period of 210 nm via *i*DME. **c** Original mask layout (Mask 0) of chip metal layer. The local layout period is smaller than the diffraction limit, i.e. $hp < \lambda/2NA$. **d** Photoresist pattern by one exposure used Mask 0. In two hotspot areas, the proximity effect is obvious and adjacent lines cannot be distinguished. **e** Two independent mask layouts (Mask 1, light green; Mask 2, light blue) of chip metal layer by disassembling the original mask in (**c**). There is no forbidden period ($hp < \lambda/2NA$) in either layout. **f** Photoresist pattern by double exposure. The adjacent lines are distinguishable because proximity effects can be avoided by multiple exposures.

## Discussion

In this study, we introduced a concept, few-photon irradiated TPA (*fp*TPA), offering a perspective on understanding the TPA process and its probability distribution under the few-photon irradiation from a tightly focused femtosecond laser pulse. The concept of *fp*TPA is based on the principles of wave-particle duality and the spatio-temporal uncertainty of photons inherent to such laser pulses. Furthermore, we have developed a spatiotemporal model to accurately describe the definite time-dependent mechanism of TPA. The simulated results using this model clearly indicate that the probability of TPA is strongly dependent on the lifetime of the molecule's virtual state under few-photon irradiation. Notably, the distribution of TPA under few-photon irradiation is significantly narrower compared to the diffraction limit of the tightly focused light spot. The results obtained from the TPA spatiotemporal model and simulations challenge existing understandings of TPA, offering a deeper insight into the TPA mechanism under few-photon irradiation and encouraging the exploration of potential applications for TPA in such conditions.

As validation, the results of TPDOPL experiments conducted using AR-N-7520 photoresist show good agreement with the simulations. Notably, by optimizing $N_{spp}$ and $N_{pulse}$ in TPDOPL, we achieved a smaller feature size of 26 nm (1/20 λ) with a laser wavelength of 517 nm, compared to 32 nm (1/12 λ) with a laser wavelength of 400 nm. Furthermore, the structure period of 210 nm (0.41 λ) and a gap distance of 37 nm was significantly decreased by performing *i*DME. This technique has proven powerful for creating dense structures when we finely control the line width. Additionally, digital projection lithography with a DMD as the digital mask is equivalent to possessing millions of individual laser focus spots, improving the patterning efficiency for multiscale structures by approximately 5 orders of magnitude. Consequently, TPDOPL under few-photon irradiation effectively breaks through the trade-off shackle between resolution and efficiency (See Supplementary Note 18 and Supplementary Table 4 for more details). We have fabricated various microstructures using diverse photoresists besides ARN 7520 to generalize the applicability of this method, such as SU-8, AR-N-4340, AR-N-5350, SCR500 and Silver/Polymer nanocomposite (see Supplementary Note 19 and Supplementary Fig. 17 for details). These results demonstrate the wide application possibilities for potential application.

The *i*DME technique in TPDOPL under few-photon irradiation is suitable not only for nanoprinting but also for nanoimaging. Although TPA microscopy has been widely applied for 3D bioimaging, its resolution has not reached the nanoscale with femtosecond laser scanning. By employing the concept of *fp*TPA and *i*DME technique, it is possible to achieve rapid imaging with nanoscale resolution. The thousands of focused spots generated by the discrete multiple focuses with DMD pattern design can simultaneously trigger TPA in thousands of molecules with minimal photon irradiation. The positions of TPA fluorescence at each focus spot can be distinctly imaged with selecting a suitable detector. By rapidly changing the designed discrete multiple focuses with DMD, a TPA fluorescence image with nanoscale resolution can be obtained in a short time.

Finally, it is noteworthy that the distribution of TPA induced by few-photon irradiation has been narrowed down to the nanometer scale, independent of the two specific wavelengths used. Theoretically, the linewidth fabricated by TPDOPL could be reduced to nearly 10 nm or less by selecting compatible photoresist molecules and optimizing processing parameters. Additionally, the minimal period would be limited only by the pixel size of the DMD with the *i*DME method. By combining TPA under few-photon irradiation with *i*DME, it is promising to achieve single-molecule imaging and nanoprinting at the sub-10 nanometer scale.

## Methods

### Experimental system and method

Using a fiber laser with a fundamental wavelength of 1035 nm, a femtosecond pulse at 517 nm was generated via a BBO crystal. The pulse repetition rate was 1 MHz, with a pulse width of 238 fs. Single-color

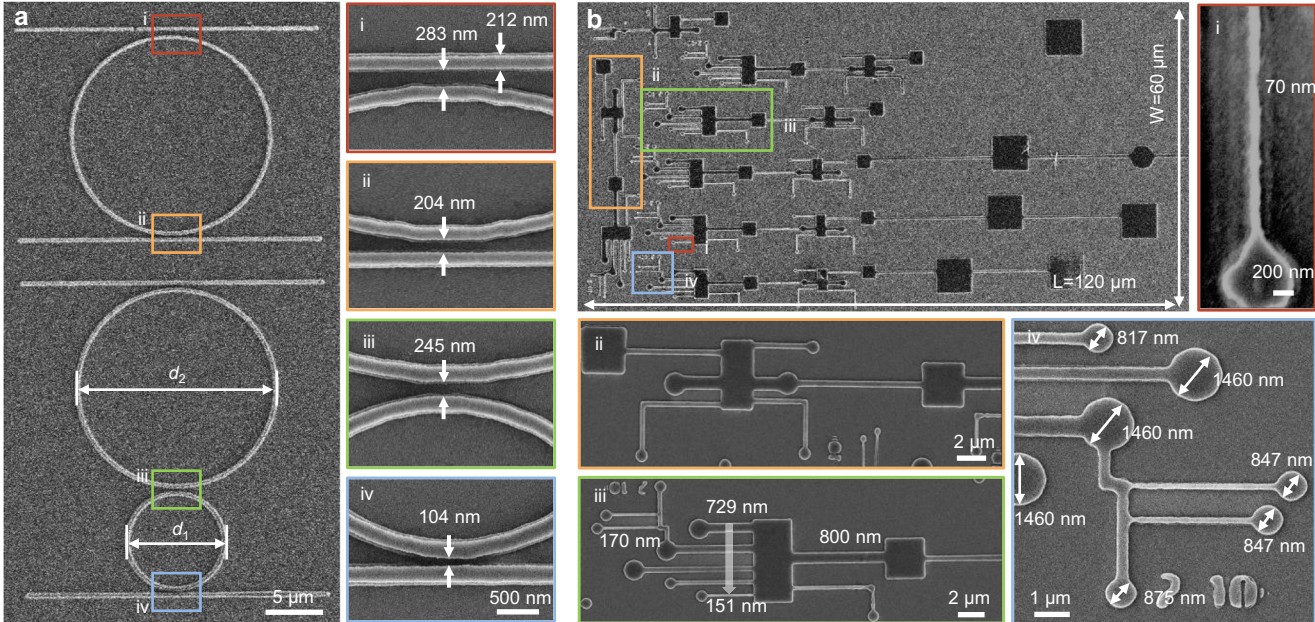

**Fig. 5 | Examples of TPDOPL with *i*DME for photonic and biological circuits.** **a** Photoresist pattern of microring waveguide with the adjustable gap at the scale of hundreds of nanometers ($d_1 = 2d_2 = 15$ μm). The insert shows several microring waveguide patterns with different gaps of 283 nm, 204 nm, 245 nm and 104 nm. **b** Photoresist pattern of biological microfluidic channels with the cross-scale feature structure. The structural feature scale covers the range of 120 μm to 70 nm. The maximum exposure structure size in a single field is $120 \times 60$ μm$^2$.

bitmap images of 1920 × 1080 pixels were created using Photoshop to meet the loading requirements of the DLP6500 1080p DMD. The images were projected onto the photoresist sample using a Nikon oil-immersion objective with 100× magnification and an NA of 1.49 (see Supplementary Note 8 for details).

Printed photoresist samples were prepared on clean glass slides (size: 24 mm × 40 mm, thickness: 0.13-0.16 mm). Hexamethyldisilazane (HMDS, adhesion promoter) spin coating was performed using an oven (model HMDS-90-M-AV) to enhance the adhesion of the subsequent photoresist. Following this, undiluted AR-N-7520 (company Allresist) commercial resist was spin-coated at 7000 rpm for 60 seconds (Spin-1200T, Midas) to achieve a uniform thin film. Subsequently, soft-baking was performed on a hotplate (NDK-1K, Asone) at 85 °C for 1 minute. After exposure, development was carried out using developer (AR 300-47, Allresist) for 1 minute at 22 °C.

### Measurement
SEM images were obtained using an Apreo 2S HiVac field-emission scanning electron microscope (FEI, company Thermo Scientific) at an acceleration voltage of 2–10 kV on a stage. Before SEM imaging, the samples were coated with a layer of Pt using a Sputter Coater (MC1000, Hitachi) to enhance its conductivity. The film thickness was measured using a step profiler (model P-7, KLA Tencor). The absorption spectra were measured using a Shimadzu UV-3600i Plus spectrophotometer. Error bars shown in all figures were calculated as the standard deviation (SD) from 10 independent measurements for each data point. Each measurement was conducted under consistent experimental conditions to minimize variability.

### Computational simulation methods
To better explore the principles of two-photon absorption under few-photon and experimentally verify them, we utilized MATLAB to establish a vector optical field distribution model based on the theory of vector optics. Subsequently, we integrated Monte Carlo random distribution algorithms to statistically distribute photons within a single pulse randomly. The statistical distribution of the reaction quantity of two-photon absorption conforms to the square of the

intensity, rendering the optical field distribution particle-like. The virtual state lifetime was obtained through the Heisenberg's uncertainty principle by using Eq. (1). The number of photons per pixel within a single pulse was calculated based on the average power behind the objective lens, while the number of pulses was determined according to the exposure time (see Supplementary Note 1–3 for details).

### Data availability
The data that supports the findings of this study are available from the corresponding authors upon request.

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

## Acknowledgements

We thank the staff of the Electron Microscopy LAB of the Institute of Photonics Technology for providing morphology characterization services. We thank the staff of the Institute of New Energy Technology for providing UV-VIS absorbance measurement services. We thank Prof. Yayi Wei and Dr. Lisong Dong from Institute of Microelectronics of Chinese Academy of Sciences for providing the layout data of the M1 metal layer. We acknowledge funding support from the National Key Research and Development Program of China (2016YFA0200500); the Major Talent Program of Guangdong Province (2019CX01Z389); the Science and Technology Planning Project of Guangzhou (202007010002); the National Natural Science Foundation of China (62005097); the Basic and

Applied Basic Research Foundation of Guangdong Province
(2023A1515010652, 2023A1515011404, 2023A1515012820).

## Author contributions

X.-M. D., M.-L. Z., Y.-Y. Z., and X.-Z. D. conceived of and designed the study. Z.-X. L., Y.-Y. Z., and X.-Z. D. set up the TPL systems, developed the control system and performed the TPL experiments. Z.-X. L., Y.-Y. Z., and J.-T. C. performed the simulations. F. J., and Z.-X. L. evaluated the photopolymer resists. Z.-X. L., and Y.-Y. Z. performed SEM imaging of nanopatterns. X.-M. D., Y.-Y. Z., M.-L. Z., Z.-X. L. and X.-Z. D. analyzed the results. X.-M. D., Z.-X. L., Y.-Y. Z. and M.-L. Z. prepared the manuscript with input from all coauthors, and all coauthors edited the manuscript.

## Competing interests

The authors declare no competing interests.
