## [Transparent Peer Review file · Nature Communications]

Two-photon absorption under few-photon irradiation for optical nanoprinting

Corresponding Author: Professor Xuan-Ming Duan

Version 0:

Reviewer comments:

Reviewer #1

(Remarks to the Author)

The manuscript “Two-photon absorption under few-photon irradiation for optical nanoprinting” by Zi-Xin Liang et al. describes the mechanisms of initiating polymerization reactions via two-photon absorption using light intensities that make statistics relevant. They perform statistical simulations considering the extremely short lifetime of the virtual intermediate during the absorption process. They then perform 2D lithography experiments to validate their simulations and mechanism. Overall, I find this to be a very interesting approach to thinking about two-photon polymerization. However, I think that the claims of beating the diffraction limit are incorrect once one considers that the two-photon sparrow criterion should be considered. The experimental setup is not new at all, many recent works have explored using a DMD for two-photon exposures. The simulations are very nice and give good insight into two-photon polymerization processes. However, in general, I think this work is not broadly impactful enough to justify publication in nature communications and would be better suited in a more specialized journal. I list some feedback for the authors below.

- The authors discuss the sparrow criterion with reference 35. Why do the authors not use the two-photon sparrow criterion as introduced in reference 35? This seems more relevant for their discussion. The authors claim to have surpassed the diffraction limit. However, the two-photon sparrow criterion says the diffraction limit should be around 123 nm for their system. Then, they have not beaten the diffraction limit.
- The authors cite reference 21 (<http://doi.org/10.1038/s41566-021-00906-8>) for their statement:

“However, with diffraction-limited focusing, the laser peak intensities can reach values as high as $I = 10^{12} \text{ W cm}^{-2}$,²¹⁻²² accompanied by corresponding photon irradiance of $3 \times 10^{31} \text{ s}^{-1} \text{ cm}^{-2}$ that is sufficient to enable appreciably effective TPA.”

The authors appear to be taking this sentence from an eerily similar sentence in the reference 21:

“On diffraction-limited focusing, peak intensities in the range of $I = 10^{12} \text{ W cm}^{-2}$ and the corresponding photon irradiance of $3 \times 10^{31} \text{ s}^{-1} \text{ cm}^{-2}$ (at a wavelength of 800 nm) provide appreciably large effective cross sections in the range of $10\text{--}19\text{--}10\text{--}20 \text{ cm}^2$, depending on the photoinitiator molecule.”

I think the authors should make it clear that they are citing this sentence as a whole and not just the intensity value as implied by the location of the citation within the sentence.

- It would be very interesting if the authors could add some discussion on a comparison between their method and the two-step absorption used in reference 21. How would long-lived real intermediate states behave in their system?
- Reference 24 (<https://doi.org/10.1126/science.aax8760>) is cited as a multi-focus approach. This is incorrect. It is basically the same approach as what the authors are presenting, just with higher pulse intensities. Further, it is unclear to me why the authors do not explain the technique of this work (or the more recent and faster version, <https://doi.org/10.1038/s41377-021-00645-z>) and how it relates to theirs. For example, the authors do not discuss if spatiotemporal focusing influences their work. Maybe it doesn't directly (?), but I would still think that the authors should address if the multiple wavelength composition of the laser pulses has an effect. All their calculations have assumed a monochromatic beam, which is not the

real case.

- The other references that are cited for multi-focus printing approaches, references 25-26, are very old. The authors should cite more recent literature such as <https://doi.org/10.37188/lam.2024.003> and <https://doi.org/10.1038/s41467-023-37163-y>
- The authors make the statement: "The throughput of TPDOPL significantly exceeds that of LDW by several orders of magnitude^{27,29}." This needs to be elaborated on and justified. What is meant by throughput? 2D patterning? 3D patterning? How is it measured?
- The authors state that the calculated intermediate state lifetimes were "confirmed". Did the authors directly measure this lifetime somehow?
- Line 166: spelling error of "pulse" in the variable subscript.
- The authors chose a step-wise polymerization photoresin for their experiments. It may be interesting if the authors could briefly elaborate if their method is at all applicable to other resist systems.
- The authors use the phrase "trade-off shackle between resolution and efficiency" several times. What explicitly do they mean by efficiency? Are they only talking about light intensity? Or do they also include scanning speed/exposure time as implied by their introduction? Some sort of figure or more direct explanation of this "shackle" would be very helpful.
- The authors spend a large part of their discussion section explaining how their method could be used for microscopy. Wouldn't there be a concern about fluorescence intensity being too low to detect with such few photon events? Then I fail to see how this method would speed up microscopy.
- Information about the tube lens used for imaging the DMD is missing.
- Line 351: What is HMDS?

Reviewer #2

(Remarks to the Author)

In the manuscript titled "Two-photon absorption under few-photon irradiation for optical nanoprinting," a new concept is presented: few-photon irradiated TPA (fpTPA), providing fresh insights into the TPA process and its probability distribution under ultralow photon irradiance using a tightly focused femtosecond laser pulse. Using fpTPA concept, a 26 nm feature size (1/20 of the wavelength) has been achieved in TPDOPL experiments.

In this work, the established concept of 'non-forgetting' photoresists was utilized, where the crosslinking degree of the monomers is determined by the exposure dose (<https://doi.org/10.1364/OE.21.010831>, <https://doi.org/10.1002/lpor.201100046>). The concept has already been applied, for example to adjust the mechanical properties of polymeric structures through multiple exposures or post-processing via multiphoton lithography, which should be recognized in this publication (i.e. <http://dx.doi.org/10.1364/OE.24.027077>, <https://doi.org/10.1116/1.345618>, DOI: 10.1039/D0NA00154F).

Nevertheless, the use of multiphoton lithography to apply this concept for enhancing feature size (and resolution) in a detailed study remains unexplored. Therefore, this contribution is worthy of a high-impact publication. The manuscript is well-written and covers two important aspects: the theoretical background of the process and the experimental work that validates the concept.

However, several points need to be addressed to further clarify and enhance the quality of the manuscript.

Major:

1.) The performed simulation does not take into account numerous relevant parameters like the refractive index of the polymeric layer, absorptivity of the photoresist etc. (some of them are already mentioned in the manuscript as being not addressed yet).

2.) Consequently, the methods employed utilize the commercially available photoresist AR-N-7520. However, specific information regarding the layer thickness are missing. In instances of thicker layers (\geq wavelength), the refractive index of the monomer may influence the focusing of the laser beam and thus also the feature size.

3.) The method has only been demonstrated for 2D lithography; this limits its applicability range.

Thus, applications highlighted by the authors, including microelectronic integrated circuits and optical waveguides, predominantly focus on 2D structures. In this regard, the method offers large-surface applicability in 2D multiphoton lithography, enabling the use of readily available low-power lasers.

The methodology is demonstrated using a single commercially available monomer, AR-N-7520, chosen for its absorption properties and directly excited and cross-linked without any modifications (no additional description provided by the authors, I assume there are no photoinitiators inside). Exploring the method's applicability to more complex photoresists would be valuable, as most 3D printing photoresists consist of a resin and photoinitiator that absorbs light (followed by for e.g. a radical formation) and triggers cross-linking reactions. In many cases, photoresists have lower viscosity, resulting in significant diffusivity of photoinitiators, radicals, and other reactive species; however, upon irradiation, the polymerization process initiates, rapidly reducing the diffusivity (by orders of magnitude) of these excited or radical-forming species (DOI: 10.1364/OE.19.019486, 10.1117/12.2287470). Reducing irradiation power, as it has been introduced in fpTPA, slows down the crosslinking process, allowing photoinitiators and radicals to diffuse more quickly, potentially increasing feature size, as observed for e.g. in STED-inspired lithography concepts (10.1002/lpor.201100046, DOI: 10.1117/12.2287470).

The major and strong limitation of this method is that it has not yet been demonstrated with other photoresists, thus restricting its general applicability to monomers that can be directly excited and applied as low-viscosity monolayers. Thus, I strongly recommend incorporating experimental data and additional paragraphs that demonstrates/address the impact of diffusion when using other low-viscosity photoresists to generalize the applicability of this method in 2D and 3D applications.

4.) What is the DMD-frequency? How fast can one switch between projection-patterns?

5.) Suppl. Mats: the absorption spectrum of the AR-N-7520 is shown to be between 250 – 500nm. Given that two-photon excitation occurs at 400 nm, I would expect the spectrum to begin at 200 nm. Please include both excitation wavelengths in the spectrum diagram. How big is the 1-photon absorbance of the AR-N-7520 @ 400 and @517nm? Since it does not look a pure 2-photon absorption, I suggest changing the title of the manuscript to multiphoton. I think an additional information on the possible impact of the 1-photon absorption for the used monomer shall be discussed (see for e.g. DOI: 10.1002/adom.202000895). I believe it will have minimal impact on fpTPA, but since the authors compare fpTPA excitation to "high-power" polymerization, addressing this might create additional value to the manuscript.

10.1002/adom.202000895). I believe it will have minimal impact on fpTPA, but since the authors compare fpTPA excitation to "high-power" polymerization, addressing this might create additional value to the manuscript.

Minor:

1.) Supplementary information p.4: please provide a graphical description for virtual state, the ground state, the intermediate state, and the transition state.

2.) fJ/pulse pixel – add brackets

3.) manuscript @ 332 : Finally, it is noteworthy that the distribution of TPA induced by few-photon irradiation has been narrowed down to the nanometer scale, independent of the light wavelength. – I believe the authors mean that the results are independent of the two wavelengths used.

4.) Manuscript @ 172: ... under few-photon irradiation can significantly can significantly ...

5.) Fig 1. Please add definitions of the letters as in Fig. 2.

6.) Please rephrase the statement; it is a bit confusing or add a sketch describing the molecular states since it is not full consistent with Fig.1 (regarding the 3 states- bring in connection to the virtual state) : ' There are three states for an active molecule in space, namely the ground state, the intermediate

state, and the transition state. The area can no longer absorb the photon in the transition state. The active molecule in the ground state will transit after absorbing one photon. To reach the intermediate state, the active molecule absorbs another photon and jumps to the transition state."

Reviewer #3

(Remarks to the Author)

Two-photon absorption under few-photon irradiation for optical nanoprinting

This work explores the concept of few-photon irradiated two-photon absorption (fpTPA) and its application for optical nanoprinting. Overall, it shows nice fabrication results, but the field of TPA/lithography is moving rapidly, and so it is important to put these results in perspective.

The concept of fpTPA is introduced as novel and potentially distinct from "traditional" two-photon absorption (TPA), but to my understanding fpTPA is just simply TPA occurring under lower photon fluxes. In practice, all practitioners in the field reduce the pulse energy so that the centre of the spatial intensity profile is just above the nonlinear polymerisation threshold. This, of course, allows for the smallest possible voxels to be created, and hence allows high-resolution 3D structuring. However, the material response is generally more important than the photon spatial and temporal distribution, as each photoresist has its own response (and hence achievable 3D resolution). I didn't see the material response discussed in detail. For example, I didn't see any mention of the chemical chain length or quenching properties. Perhaps my understanding here is limited, but I was not able to understand the novelty of fpTPA vs the standard TPA.

The authors propose a spatiotemporal model to describe TPA under few-photon irradiation. I'm not an expert in quantum, but this idea of a "real" virtual state conflicts with my understanding of the processes involved. It would be interesting to see evidence for this "real" virtual state existing in practice.

The resolution of the structures is not analysed correctly, as this requires Fourier analysis to identify resolution. See for example, Figure 3c) and e), where the "thinnest" point of the line is used. It is not a fair result to just take the thinnest line width.

The concept of using multiple lines to "get around" the diffraction limit is quite an established technique, including specifically using a DMD with multiple patterns for two-photon lithography, for achieving "sub diffraction limit" resolution. It would be good to put the novelty of your approach into perspective regarding the existing demonstrations of this approach in the literature.

"This result indicates that the resolution of TPA under few-photon irradiation can significantly surpass the diffraction limit of the employed wavelength." Again, this isn't a demonstration of surpassing the diffraction limit, in the precise definition. Diffraction limit refers to the resolvability of two point sources, not the smallest size of a feature, or the separation of two features when more than one pattern is used.

In general, the results are less impressive in resolution than others in the literature, particularly given that the results here are in 2D, and not the expected 3D that other's are demonstrating. It would be good to clarify this in more detail.

Version 1:

Reviewer comments:

Reviewer #1

(Remarks to the Author)

The authors have responded to all the points raised by the reviewers, making changes and providing strong rebuttals. I find their arguments convincing and believe that the article is suitable for publication after addressing the following minor points.

- The image quality in Figure S17m is very poor. Hard to read text.
- Scalebars in Figure S17b are illegible.
- Table S4 has different units for fabrication rate. Please make them consistent. The authors should be able to determine an area/s rate for the volume printing methods.

Reviewer #2

(Remarks to the Author)

In the revised manuscript titled "Two-photon absorption under few-photon irradiation for optical nanoprinting," improved significantly, however it requires some minor revisions:

1.) I suggest adding paper citations:

(<http://dx.doi.org/10.1364/OE.24.027077>, DOI: 10.1039/D0NA00154F) - In this context, the crosslinking degree of the monomers is controlled by the exposure dose.

Reviewer #3

(Remarks to the Author)

I am happy that the authors have addressed my concerns, and my recommendation is to accept the manuscript.

Point-to-point Response to Reviewers

Reviewer #1 (Remarks to the Author):

The manuscript “Two-photon absorption under few-photon irradiation for optical nanoprinting” by Zi-Xin Liang et al. describes the mechanisms of initiating polymerization reactions via two-photon absorption using light intensities that make statistics relevant. They perform statistical simulations considering the extremely short lifetime of the virtual intermediate during the absorption process. They then perform 2D lithography experiments to validate their simulations and mechanism. Overall, I find this to be a very interesting approach to thinking about two-photon polymerization. However, I think that the claims of beating the diffraction limit are incorrect once one considers that the two-photon Sparrow criterion should be considered. The experimental setup is not new at all, many recent works have explored using a DMD for two-photon exposures. The simulations are very nice and give good insight into two-photon polymerization processes. However, in general, I think this work is not broadly impactful enough to justify publication in nature communications and would be better suited in a more specialized journal. I list some feedback for the authors below.

Response:

We appreciate your positive comments and constructive suggestions, which would certainly help us to improve the quality of our manuscript. To address your concerns, we have carefully considered your comments and carried out additional discussions and comparative analyses to make our manuscript more convincing.

The reviewer raised two major points of contention. The first concerns the “two-photon Sparrow criterion,” which we will address and discuss in detail in the next section “**Response to comment 1**”. The second is that the experimental setup involving DMD-based projection optics combined with two-photon exposure is not novel. We agree with the reviewer’s perspective that the two-photon DMD-projection exposure scheme was first reported by Benjamin Mills’ group in 2013 [1] and improved with high performance by Shih-Chi Chen’s group in 2019 [2]. However, all these groups have employed high-power femtosecond amplified lasers with kHz pulse repetition frequency. In our previous studies [3], we conducted research on DMD-based projection exposure only using a femtosecond oscillator with MHz pulse repetition frequency. Additionally, we have investigated the relationship between the diffraction efficiency of the DMD and wavelength, which improved the performance of our DMD projection system [4]. Furthermore, we must emphasize that this paper primarily focuses on studying the photon confinement mechanism of TPA lithography under few-photon exposure, which leads to the conclusion that it is possible to achieve efficient DMD-projection two-photon lithography without using the high-power femtosecond amplified lasers. Therefore, we used the DMD-projection and two-photon exposure system to validate and demonstrate how our proposed few-photon TPA breaks the trade-off between resolution and efficiency. Rather than introducing a new optical system, we report a novel projection exposure strategy that achieves a balance between these two aspects.

Reference:

- [1] Benjamin Mills, James A Grant-Jacob, Matthias Feinaeugle, and Robert W Eason. Single-pulse multiphoton polymerization of complex structures using a digital multimirror device[J]. *Opt. Express*, **21**(12): 14853-14858 (2013).
- [2] Sourabh K. Saha, Dien Wang, Vu H. Nguyen, Yina Chang, James S. Oakdale, Shih-Chi Chen. Scalable submicrometer additive manufacturing[J]. *Science*, **366**(6461): 105-109 (2019).
- [3] Y. H. Liu et al., Multi-scale structure patterning by digital-mask projective lithography with an alterable projective scaling system. *AIP Adv.*, **8**, 065317 (2018).
- [4] M. J. Deng, Y. Y. Zhao, Z. X. Liang, J. T. Chen, Y. Zhang, X. M. Duan, Maximizing energy utilization in DMD-based projection lithography. *Opt. Express*, **30**, 4692 – 4705 (2022).

Comment 1:

• The authors discuss the sparrow criterion with reference 35. Why do the authors not use the two-photon sparrow criterion as introduced in reference 35? This seems more relevant for their discussion. The authors claim to have surpassed the diffraction limit. However, the two-photon sparrow criterion says the diffraction limit should be around 123 nm for their system. Then, they have not beaten the diffraction limit.

Response to Comment 1:

Thank you for the suggestive comments. We appreciate the opportunity to clarify this opportunity to clarify this important point. The Sparrow criterion varies depending on the experimental conditions. In our study, the system operates under parallel projection conditions, which is different from the two-photon Sparrow criterion described in serial lithography. To provide additional clarity, we have revised the relevant sections, as shown in Line 1 page 7 of manuscript.

Before discussing the sparrow criterion, it is essential to clarify whether the context involves serial or parallel lithography [1]. Traditional ultraviolet direct writing or two-photon direct writing falls under serial lithography, as shown in Fig. R1(a). When considering the two nanolines fabricated serially, the intensity distribution of the light field during two separate scanning is depicted in Fig. R2(b). It is important to emphasize that the two light fields scanned sequentially are incoherent and do not interfere with each other.

In the case of single-photon exposure, when the center distance between two scanning points is $d_{OPA} = \lambda/2NA$, the exposure doses will overlap at the center, resulting in a local minimum ($I_1(0) + I_2(0) \approx 1$). Thus, the two exposed nanolines can just be resolved, which defines the “single-photon sparrow criterion”. For two-photon exposure, where the exposure dose is proportional to the square of the intensity (I^2), when the distance between scanning points is $\lambda/2NA$, the exposure dose accumulation at the center creates a significant “valley,” making the two lines easily distinguishable (not just resolvable), as shown by the black dashed line in Fig. R1(c). For “non-forgetting” photoresists [1], the exposure dose accumulation at the center also results in a local minimum, as depicted in Fig. R1(d). At this point, $I_1(0)^2 + I_2(0)^2 \approx 1$, or $I_1(0) = I_2(0) \approx 1/\sqrt{2}$. The distance between the two line centers then becomes $d_{TPA} = d_{OPA}/\sqrt{2} = \lambda/2\sqrt{2}NA$, allowing the two nanolines to be just resolved, which defines the “two-photon sparrow criterion” [1].

Figure. R1 (a) Schematic diagram of serial scanning lithography; (b) Calculated focal intensity distribution; Lateral profiles of I (solid line) and I^2 (dotted line) correspond to one-photon exposure and two-photon exposure, respectively. (c) Critical lateral distance, d_{OPA} , for one-photon exposure; (d) Critical lateral distance, d_{TPA} , for two-photon exposure.

To ensure clarity, we have added a comparison of the Sparrow criterion for serial and parallel exposure in the Supplementary Information S13.

“In our manuscript, we employ a parallel lithography system based on digital mask projection, as illustrated in Fig. R2(a). The previously discussed two-photon sparrows criterion is applicable only to serial lithography; thus, we must now address the sparrows criterion in the context of parallel lithography, which is analogous to traditional projection lithography using physical masks [2,3]. In projection imaging systems, the sparrows criterion is established based on the condition that the optical system's modulation transfer function (MTF) equals zero [1,4]. When the line periodic resolution reaches the value defined by the sparrows criterion, the corresponding MTF condition allows for a resolvable image in the photoresist [4]. Under axial point source illumination, the cutoff frequency of the projection optical system is defined as NA/λ (where the +1 or -1 diffraction orders are situated at the edge of the entrance pupil), leading to a theoretical resolution of $d=\lambda/NA$ for lithographic imaging [5].

For single-photon exposure, the coherent linear superposition of light intensity from parallel exposure results in a local minimum appearing precisely at the center, with the exposure dose distribution of the adjacent projection light fields (proportional to intensity I) being resolvable, as shown in Fig. R2(b,d). Even with two-photon exposure, the resolvable pitch for the exposure dose between adjacent projection light fields remains at the critical value of $d=\lambda/NA$, constrained by the minimum spacing resulting from the superposition of adjacent light intensities; the exposure dose is simply proportional to I^2 . Thus, $d=\lambda/NA$ represents “the general sparrows criterion” for parallel projection lithography.

When employing off-axis point source illumination, such as through oblique illumination [3,6] or equivalent phase-shifting masks [3,7], the cutoff frequency of the projection optical system increases to $2NA/\lambda$ (which shifts the diffraction spectrum to position the +2-diffraction order at the edge of the entrance pupil), doubling the theoretical resolution to $d=\lambda/(2NA)$. Regardless of whether single-photon or two-photon exposure is utilized, the resolvable pitch for the exposure dose of adjacent projection light fields

remains at the critical value of $d=\lambda/(2NA)$, again limited by the minimum spacing from adjacent light intensity superposition, as depicted in Fig. R2(c,e). Consequently, $d=\lambda/(2NA)$ defines the sparrow criterion for parallel projection lithography under off-axis illumination conditions.

Figure. R2 (a) Schematic diagram of parallel projection lithography; (b,c) Calculated focal intensity distribution for single exposure under off-axis illumination and off-axis illumination (or phase shift mask), respectively; Lateral profiles of I (solid line) and I^2 (dotted line) correspond to one-photon exposure and two-photon exposure, respectively. (d,e) Critical lateral distance for one/two-photon single exposure under off-axis illumination and off-axis illumination (or phase shift mask), respectively.

The digital mask projection lithography system discussed in this manuscript uses a collimated ultrafast laser beam aligned parallel to the optical axis, equivalent to the illumination of an object point at infinity on-axis. This setup corresponds to “the general sparrow criterion” for parallel projection lithography, where $d=\lambda/NA$. In our experiments, we used a laser with a center wavelength of 517 nm and an objective lens with an NA of 1.49, yielding a theoretical line pitch resolution of approximately 347 nm. Thus, a single exposure cannot achieve a line array pattern with period of 227 nm (corresponding to 3 pixels). However, by using double exposure (with a single-exposure pattern resolution of 454 nm, equivalent to 6 pixels, which adheres to the sparrow criterion), we obtained clear, distinguishable exposure results, as shown in Fig. 4 of the manuscript.” To avoid ambiguity, we have revised the manuscript’s descriptions (Line 25 Page 7).

Reference:

- [1] Fischer J, Wegener M. Three-dimensional optical laser lithography beyond the diffraction limit[J]. *Laser Photonics Rev.*, **7**(1): 22-44 (2013).
- [2] Andreas Erdmann. Optical and EUV Lithography: A Modeling Perspective[M]. Publisher. SPIE; Publication date. March 2 (2021).
- [3] Schellenberg F M. Resolution enhancement technology: the past, the present, and extensions for the future[C]//Optical Microlithography XVII. SPIE, 5377: 1-20 (2004).
- [4] Sparrow C M. On spectroscopic resolving power[J]. *Astrophysical Journal*, **44**:76, (1916).
- [5] Masters B R, Masters B R. Abbe’s theory of image formation in the microscope[J]. *Superresolution optical microscopy: The quest for enhanced resolution and contrast*, 65-

108 (2020).

[6] Shibuya M. Resolution enhancement techniques for optical lithography and optical imaging theory[J]. *Opt. Rev.*, **4**: 151-160 (1997).

[7] Levenson M D, Viswanathan N S, Simpson R A. Improving resolution in photolithography with a phase-shifting mask[J]. *IEEE Trans. Electron Devices*, **29**(12): 1828-1836 (1982).

Comment 2:

- The authors cite reference 21 (<http://doi.org/10.1038/s41566-021-00906-8>) for their statement:

“However, with diffraction-limited focusing, the laser peak intensities can reach values as high as $I = 10^{12} \text{ W cm}^{-2}$,²¹⁻²² accompanied by corresponding photon irradiance of $3 \times 10^{31} \text{ s}^{-1} \text{ cm}^{-2}$ that is sufficient to enable appreciably effective TPA.”

The authors appear to be taking this sentence from an eerily similar sentence in the reference 21:

“On diffraction-limited focusing, peak intensities in the range of $I = 10^{12} \text{ W cm}^{-2}$ and the corresponding photon irradiance of $3 \times 10^{31} \text{ s}^{-1} \text{ cm}^{-2}$ (at a wavelength of 800 nm) provide appreciably large effective cross sections in the range of 10^{-19} – 10^{-20} cm^2 , depending on the photoinitiator molecule.”

I think the authors should make it clear that they are citing this sentence as a whole and not just the intensity value as implied by the location of the citation within the sentence.

Response to Comment 2:

Thank you for your suggestion. We should have been more explicit in citing the entire sentence rather than just the intensity value. We apologize for this oversight and appreciate your diligence in ensuring academic rigor. In response to your feedback, we have revised the sentences and cited the references in the revised manuscript to accurately reflect our respect for the original work and to address your concern appropriately.

The revised sentence is on **Page 2 (Line 17)** in the revised manuscript: “However, with diffraction-limited focusing, the laser peak intensities can reach values as high as $I = 10^{12} \text{ W cm}^{-2}$, accompanied by corresponding photon irradiance of $3 \times 10^{31} \text{ s}^{-1} \text{ cm}^{-2}$ that is sufficient to enable appreciably effective TPA²¹⁻²².”

Comment 3:

- It would be very interesting if the authors could add some discussion on a comparison between their method and the two-step absorption used in reference 21. How would long-lived real intermediate states behave in their system?

Response to Comment 3:

Thank you for your insightful comments. This comment has drawn our attention to the comparison between our work and the innovative approach presented in reference 21, which introduces a two-step absorption (TSA) mechanism for 3D nonprinting.

The two methods have different mechanisms. Our study introduces a novel concept of few-photon irradiated two-photon absorption (*fp*TPA), which offers a distinct perspective on the two-photon absorption process under ultralow photon irradiance conditions. By leveraging the wave-particle duality of light and photon distribution, we describe the

probability and distribution of effective TPA (eTPA) as a function of photon irradiance. The probability of two-photon absorption is proportional to the time correlation function of the two photons in the virtual state lifetime. Thus, the simulated autocorrelation number N_{TPA} is the integration of the time correlation of time pulse function intensity as $N_{TPA} = \int n(t) \cdot n(t-t_0) dt_0$ (t_0 integral range 0- τ), τ is the lifetime of the virtual level (Generally in the order of 1 fs). Therefore, the number of two-photon absorptions is nonlinearly dependent on the light intensity or the number of photons.

Compared to two-photon absorption, two-step absorption (TSA) relies on real intermediate states (real energy levels), which have a very long lifetime, generally reaching a ps even ns levels. The first excited state acts as a long-lived intermediate state to cascade two single-photon absorption processes, which ultimately leads to the number of absorbed photons N_{TSA} being proportional to I^2 . Thus, two kinds of excited states, introduced by the normal single photon absorption and the two-step absorption, should be involved. The lifetime of the intermediate state directly influences the overall conversion efficiency. The absorption efficiency increases when the time window for the absorption of the second photon is extended. However, we can't confirm that what percentage of the polymerization is introduced by the two-step absorption, since there are different relaxing mechanisms for the different excited states, the normal single photon absorption is also possible introducing a polymerization. A similar effect is observed in our two-photon absorption process: a longer virtual state lifetime enhances the probability of the second photon being absorbed. As shown in **Section S4** and **Fig. S3** of the Supporting Information, under identical conditions for other influencing parameters, different wavelengths result in varying virtual state lifetimes. This variation affects the effective two-photon absorption (N_{eTPA}). For example, at 400 nm, the longer virtual state lifetime leads to a higher probability of effective two-photon absorption events.

Comment 4:

• Reference 24 (<https://doi.org/10.1126/science.aax8760>) is cited as a multi-focus approach. This is incorrect. It is basically the same approach as what the authors are presenting, just with higher pulse intensities. Further, it is unclear to me why the authors do not explain the technique of this work (or the more recent and faster version, <https://doi.org/10.1038/s41377-021-00645-z>) and how it relates to theirs. For example, the authors do not discuss if spatiotemporal focusing influences their work. Maybe it doesn't directly (?), but I would still think that the authors should address if the multiple wavelength composition of the laser pulses has an effect. All their calculations have assumed a monochromatic beam, which is not the real case.

Response to Comment 4:

Thank you for your comment. We have revised the incorrect description into "Although multi-focus²⁴⁻²⁷ techniques can partially enhance fabrication efficiency, the serial point-by-point writing protocol of LDW remains inadequate for efficiently fabricating structures with multiscale components, ranging from nanoscale to macroscale" on **Page 2 (Line 23)** in the revised manuscript.

Regarding the relevance of this study to related technologies, this was not highlighted in the background due to differences in motivation. Our manuscript primarily focuses on

developing a spatiotemporal model of the two-photon process from a photonic perspective, to understand photon-material interactions under few photons' irradiation. This approach aims to improve two-photon lithography resolution under extremely low light conditions. By integrating this method with an optical projection exposure optical system, we achieve high-resolution, high-efficiency patterning of photoresist using a femtosecond oscillator with low pulse energy ($\sim\mu\text{J}$ or sub- μJ per pulse) without relying on a high-power femtosecond amplifier ($\sim\text{mJ}$ per pulse). To demonstrate the distinct advantages of our proposed method, we have summarized and compared the technical parameters of our work with those of related studies in a comparative table, which is included in the S18 in Supporting Information.

Due to the periodic micro-mirror array structure of the DMD, diffraction inevitably occurs upon coherent light hitting it. When using a broadband laser source, the diffraction angles of light beams with different wavelengths slightly vary, resulting in spectral separation. This separation leads to the stretching of pulse duration, which reduces the intensity gradient at the focal plane of the objective lens and consequently affects polymerization within the thin layer along the Z-direction. Reference 24 employs femtosecond pulses with a broad wavelength spectrum (approximately ten of nanometers) and addresses this issue by introducing the temporal focusing technique. This approach ensures that the pulse is recompressed to its shortest duration exclusively at the spatial focal plane of the objective lens.

In contrast, the spatiotemporal two-photon polymerization mechanism we propose fundamentally differs from the aforementioned approach. Our study focuses on the intrinsic temporal and spatial mechanisms involved in the absorption of two photons by the same molecular entity, specifically analyzing the reaction probability of two-photon absorption (TPA) processes within the virtual state lifetime.

In our study, a fiber femtosecond laser with a narrow spectral bandwidth is employed, and the thickness of the photoresist film is reduced to mitigate the impact of Z-axis effects on the accuracy of the model. We used a femtosecond light source with a center wavelength of 517 nm, pulse width of ~ 240 fs, and bandwidth (FWHM) of approximately 3 nm shown in Fig. R2, while the cited literature used a source with an 800 nm center wavelength, ~ 35 fs pulse width, and 40 nm bandwidth. Subsequently, in 2021, Xianfan Xu et al. [2] employed DMD projection processing with a laser amplifier characterized by a central wavelength of 800 nm, a bandwidth of 22 nm, and a pulse duration of 65 fs. Given the broad bandwidth of the laser amplifiers in both studies, it was essential to account for the impact of non-monochromatic light. As a result, each study incorporated dispersion compensation to address the chromatic dispersion effects associated with polychromatic light. Based on the calculated focus spot morphology for different bandwidths (see Fig. S6 in SI) and reference [3,4], the impact of spatiotemporal focusing on spot morphology is minimal and can be disregarded in our study.

Fig. R2 Spectrum of femtosecond light source used in this study

Reference:

- [1] Saha S K, Wang D, Nguyen V H, et al. Scalable submicrometer additive manufacturing[J]. *Science*, **366**(6461): 105-109 (2019).
- [2] Somers P, Liang Z, Johnson J E, et al. Rapid, continuous projection multi-photon 3D printing enabled by spatiotemporal focusing of femtosecond pulses[J]. *Light Sci. Appl.*, **10**(1): 199 (2021).
- [3] He F, Xu H, Cheng Y, et al. Fabrication of microfluidic channels with a circular cross section using spatiotemporally focused femtosecond laser pulses[J]. *Opt. Lett.*, **35**(7): 1106-1108 (2010).
- [4] Cheng Y, Xie H, Wang Z, et al. Onset of nonlinear self-focusing of femtosecond laser pulses in air: Conventional vs spatiotemporal focusing[J]. *Phys. Rev. A.*, **92**(2): 023854 (2015).

Comment 5:

- The other references that are cited for multi-focus printing approaches, references 25-26, are very old. The authors should cite more recent literature such as <https://doi.org/10.37188/lam.2024.003> and <https://doi.org/10.1038/s41467-023-37163-y>

Response to Comment 5:

Thanks to the comment. We have updated the references, including the references provided by the reviewer (references 24-27).

Comment 6:

- The authors make the statement: "The throughput of TPDOPL significantly exceeds that of LDW by several orders of magnitude^{27,29}." This needs to be elaborated on and justified. What is meant by throughput? 2D patterning? 3D patterning? How is it measured?

Response to Comment 6:

Thank you for the suggestion. We have revised the supplementary to clarify the meaning and justification of "throughput" (S10).

“Throughput” is an important parameter for characterizing lithography efficiency, typically representing the area or volume of patterned structures produced per unit of time. The definition of throughput slightly varies across different lithography technologies, for example,

1. In traditional mask-based lithography, “throughput” is generally defined as the number of wafers processed per hour, with units in wafers per hour (WPH) [1]. For example, an EUV lithography machine using 12-inch wafers (300 mm) has a throughput of 170 WPH.

2. For 2D patterning, in processes such as electron beam lithography, laser direct-write lithography, or maskless UV lithography [2], “throughput” is defined as the area exposed per minute, with units in mm²/min or mm²/s.

3. For 3D patterning, in two-photon lithography [3], since the smallest exposure unit is a voxel, “throughput” is defined as the number of voxels exposed per second, with units in Voxels/s. Once the voxel size is determined, it can be converted into mm²/min or mm²/s. In UV-cured 3D printing [4], “throughput” is defined as the volume exposed per second, with units in mm³/s.

In our research, we focus on 2D patterning of photoresists, where “throughput” is defined as the area exposed per minute, with units in mm²/min. We have calculated the throughput values in our experimental process and updated them in the manuscript (see Line 31 Page 7). The method for calculating throughput is equal to the exposed field area per exposure / effective exposure time per field. Factors influencing throughput include the DMD target area, objective lens magnification, DMD refresh rate, and the sensitivity of the photoresist, among others. This method can also be extended to 3D lithography.

Reference:

[1] TWINSKAN NXE:3400C – EUV lithography systems | ASML

<https://www.asml.com/en/products/euv-lithography-systems/twinscan-nxe3400c>

[2] Kang M, Han C, Jeon H. Submicrometer-scale pattern generation via maskless digital photolithography[J]. *Optica*, **7**(12): 1788-1795 (2020).

[3] Somers P, Koch S, Kiefer P, et al. Holographic multi-photon 3D laser nanoprinting—at the speed of light: opinion[J]. *Opt. Mater. Express*, **14**(10): 2370-2376 (2024).

[4] Ge Q, Li Z, Wang Z, et al. Projection micro stereolithography based 3D printing and its applications[J]. *Int. J. Extreme Manuf.*, **2**(2): 022004(2020).

Comment 7:

• The authors state that the calculated intermediate state lifetimes were “confirmed”. Did the authors directly measure this lifetime somehow?

Response to Comment 7:

Thank you for your comment. The virtual state lifetime discussed in the manuscript is not directly measured; rather, it is estimated based on the uncertainty principle in quantum mechanics. The specific details are as follows:

The Heisenberg energy-time uncertainty tells us that we can have so-called virtual states between eigenstates as long as the lifetime of these states at most:

$$\tau = h/(4\pi E_v)$$

Where h is the Planck constant ($6.62607 \times 10^{-34} \text{ J}\cdot\text{s}$, or $4.135668 \times 10^{-15} \text{ eV}\cdot\text{s}$), and E_v is

energy difference between our virtual state and the nearest eigenstate. If we assume $E_v \approx 1\text{eV}$ for a virtual state, this state can be estimated only exist for at most $\tau \approx 3.3 \times 10^{-16}$ seconds ≈ 0.33 fs.

In quantum mechanics, a virtual state is an extremely transient and unobservable quantum state. In quantum processes such as multiphoton absorption or multiphoton ionization, the virtual state acts as an intermediate state in electron transitions and is described as "imaginary" [1]. Since a virtual state is not an eigenfunction of any operator, physically observable quantities are represented by self-adjoint operators, and thus, the "virtual state" is considered an unmeasurable state [2]. Conventional parameters like occupancy, energy, and lifetime require careful clarification. While a measurement of a system would not reveal occupancy of a virtual state, its lifetime can still be inferred from the uncertainty principle. Although each virtual state has an associated energy, its energy cannot be measured directly [3]. However, various methods have been used to obtain certain measurements (see related work on virtual state spectroscopy) or to extract other parameters using techniques that depend on the virtual state lifetime [4-5]. One might assume that it is only the sensitivity threshold of current instruments that limits the measurement of quantities associated with virtual states, suggesting that the intermediate state might be theoretically detectable in an experiment, but practical constraints make this difficult [3]. However, the opposite is true: regardless of the resolution capability of measuring equipment, the energy of a virtual state is fundamentally unmeasurable. This can be explained through Heisenberg's uncertainty principle, which implies that assuming the virtual state at an extremely short lifetime $\Delta t \approx 0$ is reasonable. This would result in an infinitely large energy uncertainty, which clearly defies physical principles since the energy of a virtual state lies between the excited and ground states.

Reference:

- [1] Arthur L. Robinson, Tunable Far IR Molecular Lasers Developed. *Science*, **227**:736-737 (1985).
- [2] Masters BR. "Historical Development of Non-linear Optical Microscopy and Spectroscopy". In Masters BR, So P (eds.). Handbook of Biomedical Nonlinear Optical Microscopy[M]. US: Oxford University Press (2008).
- [3] Belkic D. Principles of quantum scattering theory[M]. CRC Press (2020).
- [4] Saleh B E A, Jost B M, Fei H B, et al. Entangled-photon virtual-state spectroscopy[J]. *Phys. Rev. Lett.*, **80**(16): 3483 (1998).
- [5] Kojima J, Nguyen Q V. Entangled biphoton virtual-state spectroscopy of the $A^2\Sigma^+ - X^2\Pi$ system of OH[J]. *Chem. Phys. Lett.*, **396**(4-6): 323-328 (2004).

Comment 8:

- Line 166: spelling error of "pulse" in the variable subscript.

Response to Comment 8:

Thank you for your comment. We have checked the whole manuscript and corrected some spelling errors.

Comment 9:

- The authors chose a step-wise polymerization photoresin for their experiments. It may be interesting if the authors could briefly elaborate if their method is at all applicable to other resist systems.

Response to Comment 9:

Thank you for your comment, which would help us improve the quality of our manuscript. As suggested, we have summarized the fabrication process using various other photoresists. In fact, our method have been successfully used to various photoresist systems, such as chemical amplifier negative photoresist (AR-N-4340 and SU-8), non-chemical amplifier positive photoresist (AR-N-5350), free-radical photoresist (SCR500) and Silver/Polymer nanocomposite with low viscosity.

Moreover, we have added a corresponding discussion on Page 9 (Line 14) in the revised manuscript: "We have fabricated various microstructures using diverse photoresists besides ARN 7520 to generalize the applicability of this method, such as SU-8, AR-N-4340, AR-N-5350, SCR500 and Silver/Polymer nanocomposite (see Supplementary S19 for details). These results demonstrate the wide application possibilities for potential application".

Comment 10:

- The authors use the phrase "trade-off shackle between resolution and efficiency" several times. What explicitly do they mean by efficiency? Are they only talking about light intensity? Or do they also include scanning speed/exposure time as implied by their introduction? Some sort of figure or more direct explanation of this "shackle" would be very helpful.

Response to Comment 10:

Thank you for your suggestion. Both "efficiency" and the previously mentioned "throughput" refer to the production capacity of lithography technology, indicating the maximum pattern area or number of wafers that can be exposed per unit time. A higher throughput implies greater efficiency.

As noted by the reviewer, the intuitive experimental conditions that influence "efficiency" include light intensity, scanning speed/exposure time, objective field of view, and the sensitivity of the photoresist. In contrast to traditional point-scanning TPA lithography (where point-scanning exposure is generally considered a serial, low-efficiency lithography method), this manuscript focuses on controlling the spatially confined photochemical reaction process between photons and materials under low light intensity to achieve enhanced resolution in two-photon lithography under extremely low light conditions. This approach is combined with an optical projection exposure optical system (generally considered a parallel, high-efficiency lithography method), enabling high resolution and high efficiency without requiring a high-power femtosecond amplifier (single pulse energy in the \sim mJ range). Instead, by using a femtosecond oscillator with lower single pulse energy (\sim μ J or sub- μ J range), we are able to achieve high resolution and high efficiency using a few-photon TPA lithography technique.

To explain how our proposed method overcomes the "bottleneck" of balancing resolution and efficiency, we calculated the throughput values representing "efficiency" and provided a detailed comparison with similar methods and their relevant parameters (as shown in the table below and included in Supporting Information S18). The table shows

that our proposed method achieves **high levels in both linewidth resolution and line spacing resolution, and its efficiency is on the same order of magnitude as similar methods, approximately 10^6 voxels/s.** For traditional scanning-based techniques, achieving higher resolution requires smaller voxel volumes, which naturally reduces the area covered in a single exposure. As a result, printing the same area with higher resolution demands significantly more time. The throughput of TPL remains largely unaffected by the number of voxels per layer due to its capability to simultaneously project an entire 2D layer [2]. Our method employs a projection exposure system that efficiently balances resolution and throughput. Large-scale pattern exposure at the scale of 100 microns can be achieved in a single exposure under the two-photon optical projection system, achieving a fabrication rate of 1×10^{-3} mm²/s. Under exposure with the DLP 9000 (2560 × 1600 pixels, 7.56 μm pixel pitch, Texas Instruments) and a 40x objective lens (Fluar, 1.30 Oil, Zeiss), a single exposure area of 250 μm × 400 μm can be achieved, realizing a fabrication rate of 0.1 mm²/s (4×10^6 voxels/s). We have also supplemented the information in S10 of the supplementary.

Year Group	Method		Source Power density	Resolution	Throughput	
	Principle	Type			Fabrication rate	Voxel fabrication rate (v_{voxels}) (voxels/s)
2013 ^[1] Benjamin Mills	MPP	DMD projection	Femtosecond amplified laser, 800 nm, 150 fs, 1 kHz; Single pulse 1 mJ	Width: 650 nm (0.45λ/NA); Pitch: no data	0.5mm ² /min (8.3×10^{-3} mm ² /s)	No data
2019 ^[2] Shih-Chi Chen	TPP	DMD projection	Femtosecond amplified laser, 800 nm, 35fs, ~40 nm bandwidth, 1 kHz, P~1W; 1 TW/cm ²	Lateral widths: 130-140 nm (~0.21λ/NA); Axial height: 175 nm Pitch: no data /	2D: ~9 mm ² /s 3D: 5-20mm ³ /hour (0.83-3.33) × 10 ⁻² mm ³ /s	3.33×10^8
2021 ^[3] Xianfan Xu	TPP	DMD projection+ spatiotemporal focusing	Femtosecond amplified laser, 800 nm, 65 fs, 22 nm bandwidth; 5 kHz 0.44 TW/cm ²	Width: ~200 nm (0.37λ/NA); Layer thickness: ~1 μm Pitch: no data	~10 ⁻³ mm ³ /s	3.2×10^6
2021 ^[4] Xuan-Ming Duan	TPP	DMD projection	Femtosecond laser oscillator, 400 nm, 100 fs, 80 MHz, P~1W; 0.14-0.365 MW/cm ²	Width: 32 nm (0.12λ/NA); Pitch: no data	1.74×10^{-2} mm ² /s	~1×10 ⁶

2022 ^[5] Martin Wegener	TSA+LCD	Light-sheet	Four 6 W optical-power, 440 nm wavelength laser diodes $I_1 = 162 \mu\text{W}/\mu\text{m}^2$ (0.0162 MW/cm ²)	Width: 500 nm; Minimum ~250 nm (0.68 λ /NA) Pitch: no data	3.85 $\times 10^6 \mu\text{m}^3/\text{s}$ (3.85 $\times 10^{-3} \text{mm}^3/\text{s}$)	7 $\times 10^6$
2023 ^[6] Shih-Chi Chen	TPP	DMD Holography multi-foci	Femtosecond amplified laser, 800 nm, 100fs, 1kHz, P=4 W; 3.3~22.7 TW/cm ²	Width: 90 nm (0.15 λ /NA); Pitch: ~500 nm (0.81 λ /NA)	54 mm ³ /hour (1.5 $\times 10^{-3} \text{mm}^3/\text{s}$)	2 $\times 10^6$
2024 ^[7] Martin Wegener	TPP	DOE multi-foci (7 \times 7)	Femtosecond laser oscillator, 790nm, 140fs, 80MHz, P=3.7W; 1.2TW/cm ² per focus	Mean lateral size: 475 nm (0.84 λ /NA) Pitch: no data	3.6 $\times 10^{-3} \text{mm}^3/\text{s}$	10 ⁸
This work	TPP	DMD projection	Fiber Femtosecond Laser, 517nm, ~240 fs, 1 MHz; 0.17 GW/cm²	Minimum width: 26 nm (0.075λ/NA); Pitch: 210 nm (0.6λ/NA)	~1$\times 10^{-3} \text{mm}^2/\text{s}$(26nm) ~0.1 mm²/s(250nm)	~4$\times 10^6$

Reference:

- [1] Benjamin Mills, James A Grant-Jacob, Matthias Feinaeugle, and Robert W Eason. Single-pulse multiphoton polymerization of complex structures using a digital multimirror device[J]. *Opt. Express*, **21**(12): 14853-14858 (2013).
- [2] Sourabh K. Saha, Dien Wang, Vu H. Nguyen, Yina Chang, James S. Oakdale, Shih-Chi Chen. Scalable submicrometer additive manufacturing[J]. *Science*, **366**(6461): 105-109 (2019).
- [3] Paul Somers, Zihao Liang, Jason E. Johnson, Bryan W. Boudouris, Liang Pan, and Xianfan Xu. Rapid, continuous projection multi-photon 3D printing enabled by spatiotemporal focusing of femtosecond pulses[J]. *Light Sci. Appl.*, **10**(1): 199 (2021).
- [4] Yu-Huan Liu, Yuan-Yuan Zhao, Feng Jin, Xian-Zi Dong, Mei-Ling Zheng, Zhen-Sheng Zhao, and Xuan-Ming Duan. $\lambda/12$ super resolution achieved in maskless optical projection nanolithography for efficient cross-scale patterning[J]. *Nano Lett.*, **21**(9): 3915-3921(2021).
- [5] Vincent Hahn, Pascal Rietz, Frank Hermann, Patrick Müller, Christopher Barner-Kowollik, Tobias Schlöder, Wolfgang Wenzel, Eva Blasco, and Martin Wegener. Light-sheet 3D microprinting via two-colour two-step absorption[J]. *Nat. Photonics*, **16**(11): 784-791 (2022).
- [6] Wenqi Ouyang, Xiayi Xu, Wanping Lu, Ni Zhao, Fei Han, and Shih-Chi Chen. Ultrafast 3D nanofabrication via digital holography[J]. *Nat. Commun.*, **14**(1): 1716(2023).

[7] Pascal Kiefer, Vincent Hahn, Sebastian Kalt, Qing Sun, Yolita M. Eggeler, Martin Wegener. A multi-photon (7×7)-focus 3D laser printer based on a 3D-printed diffractive optical element and a 3D-printed multi-lens array[J]. *Light: Advanced Manufacturing*, 4(1): 28-41(2024).

Comment 11:

• The authors spend a large part of their discussion section explaining how their method could be used for microscopy. Wouldn't there be a concern about fluorescence intensity being too low to detect with such few photon events? Then I fail to see how this method would speed up microscopy.

Response to Comment 11:

Thank you for your comment. You raise a valid concern regarding the potential challenge of low fluorescence intensity when detecting few photon events. Our method leverages the capabilities of the DMD (Digital Micromirror Device) to control millions of micromirrors in parallel, enabling the precise light illumination across large sample areas or providing random pixel control for multifocal illumination. The key to overcoming the low photon count issue is the use of multiple illumination pulses in the single-pixel approach, where light from multiple pulses is accumulated to improve the signal-to-noise ratio, even when photon flux is low. This technique ensures that, despite the reduced number of photons per event, the signal can be enhanced through temporal averaging, making it easier to detect. Additionally, the DMD's high refresh rate (up to 22.7 kHz) allows for rapid switching and control of the light source, significantly speeding up the image acquisition process. While each individual photon event may have low intensity, the ability to rapidly accumulate high-quality data through temporal and spatial multiplexing helps accelerate microscopy, enabling faster image capture while maintaining signal sensitivity.

Existing super-resolution fluorescence microscopy techniques are limited by trade-offs between spatial resolution, temporal resolution, and field of view, making it challenging to achieve large-field, high-frame-rate, super-resolution imaging. This manuscript proposes a few-photon-irradiation-based two-photon absorption mechanism, where a digital micromirror device (DMD) is used to pattern-modulate few-photon pulse light. The patterned light is then projected through an optical reduction imaging objective system onto a sample containing dispersed fluorescent molecules for two-photon excitation. High spatiotemporal resolution detection of single-molecule fluorescence is achieved using detectors such as single-photon counters, EMCCD, high-sensitivity spectrometers, and photomultiplier tubes (PMTs), capturing the spatial and temporal information of two-photon excitation and fluorescence. We have also revised the description in manuscript, as shown on page 9, line 28.

By designing pixelated patterns on the DMD, a large number of discrete excitation points with inter-point distances exceeding the optical diffraction limit are created using single-photon/few-photon pulse excitations. This setup allows for the acquisition of two-photon fluorescence distributions of single molecules, providing experimental data to study the quantum spatiotemporal statistical patterns of photon-molecule interactions. When applied to biological samples, such as live cells, the rapid switching of discrete excitation points across the field enables high spatiotemporal resolution imaging of biological

dynamics.

Comment 12:

- Information about the tube lens used for imaging the DMD is missing.

Response to Comment 12:

Thank you for pointing out the missing information regarding the tube lens used for imaging the Digital Micromirror Device (DMD). The details are essential for the reproducibility of our results and for a better understanding of our optical setup. We have included additional relevant information in supplementary (S8): The optical system can project any binarized pattern loaded onto the DMD onto the target photoresist through the tube mirror (TTL200-UVB, Thorlabs) and the objective lens (Apo TIRF, NIKON, oil, 100× /1.49 NA, Nikon) without moving the stage (Fig. S7A).

Comment 13:

- Line 351: What is HMDS?

Response to Comment 13:

HMDS (Hexamethyldisilazane) is an adhesion promoter commonly used in photolithography. It forms a monomolecular layer on the substrate surface, significantly enhancing the adhesion between the photoresist and materials such as metals, SiO₂, and GaAs. This improves the integrity and stability of photoresist patterns during exposure and development, which is crucial for achieving high-resolution patterns [1]. In our study, the use of HMDS demonstrated a marked improvement in the quality of the photoresist patterns, supporting more precise and reliable photolithography. Additionally, we have added the corresponding description on page 10 line 8 of the manuscript to facilitate understanding for the readers.

[1] Manako S, Ochiai Y, Fujita J, et al. Nanolithography using a chemically amplified negative resist by electron beam exposure[J]. Japanese Journal of Applied Physics, **33**(12S): 6993 (1994).

Reviewer #2 (Remarks to the Author):

In the manuscript titled "Two-photon absorption under few-photon irradiation for optical nanoprinting," a new concept is presented: few-photon irradiated TPA (fpTPA), providing fresh insights into the TPA process and its probability distribution under ultralow photon irradiance using a tightly focused femtosecond laser pulse. Using fpTPA concept, a 26 nm feature size (1/20 of the wavelength) has been achieved in TPDOPL experiments.

In this work, the established concept of 'non-forgetting' photoresists was utilized, where the crosslinking degree of the monomers is determined by the exposure dose (<https://doi.org/10.1364/OE.21.010831>, <https://doi.org/10.1002/lpor.201100046>). The concept has already been applied, for example to adjust the mechanical properties of polymeric structures through multiple exposures or post-processing via multiphoton lithography, which should be recognized in this publication (i.e. <http://dx.doi.org/10.1364/OE.24.027077>, <https://doi.org/10.1116/1.345618>, DOI: 10.1039/D0NA00154F).

Nevertheless, the use of multiphoton lithography to apply this concept for enhancing feature size (and resolution) in a detailed study remains unexplored. Therefore, this contribution is worthy of a high-impact publication. The manuscript is well-written and covers two important aspects: the theoretical background of the process and the experimental work that validates the concept.

However, several points need to be addressed to further clarify and enhance the quality of the manuscript.

Response:

We appreciate your positive comments and constructive suggestions, which would certainly help us to improve the quality of our manuscript. We have carefully considered your comments and carried out more experiments to make our manuscript more convincing.

Major:

Comment 1:

The performed simulation does not take into account numerous relevant parameters like the refractive index of the polymeric layer, absorptivity of the photoresist etc. (some of them are already mentioned in the manuscript as being not addressed yet).

Response:

Thank you for your insightful comment. We acknowledge that several parameters considered in our simulations, such as the refractive index of the polymeric layer and the absorptivity of the photoresist, were not included in the Supplementary Information (SI) file, which was an oversight on our part. We have now added this data in Table R2 located in section S3 of the SI.

Regarding the absorptivity of the photoresist, we recognize that it could lead to pattern distortion. However, since the absorptivity depends on various factors such as the concentration and thickness of the photoresist, the actual effect may vary with different resists. Therefore, we have considered reducing the film thickness in our experiments to minimize the impact of the Z-direction height on the effective reaction distribution. For the theoretical calculations, we have chosen to temporarily exclude the specific value of absorptivity in this study.

Table R2 Numerical values of parameters in the optical model for few-photon absorption.

Parameter	Numerical	Numerical	Units
Cutoff Line	353	353	nm
Numerical Aperture (NA)	1.45	1.49	Non-dimensional
Wavelength (λ)	400	517	nm
Virtual life	0.7973	0.2953	fs
FWHM	100	238	fs
magnification	90	100	Non-dimensional
refractive_index (photoresist)	1.623	1.622	Non-dimensional
DMD_size	7.56	7.56	μm

Comment 2:

Consequently, the methods employed utilize the commercially available photoresist AR-N-7520. However, specific information regarding the layer thickness are missing. In instances of thicker layers (\geq wavelength), the refractive index of the monomer may influence the focusing of the laser beam and thus also the feature size.

Response:

Thank you for your valuable comment. We agree that the thickness of the photoresist layer can impact the resulting structure. Due to the coherence of the laser, the interference between incident and reflected light within the photoresist can create standing waves, which may affect the patterned structures [1].

The refractive index of the photoresist was calculated using the Cauchy coefficients provided in the commercial data sheet for AR-N-7520. According to these coefficients ($N_0 = 1.622, N_1 = 123.2,$ and $N_2 = 0$), we applied the Cauchy equation:

$$n(\lambda) = N_0 + \frac{N_1}{\lambda^2} + \frac{N_2}{\lambda^4}$$

to determine the refractive index at a wavelength of 517 nm, yielding $n(517) \approx 1.6225$.

Using the formula for the minimum thickness unaffected by standing waves, $d_{min} = \frac{\lambda}{2n}$, we obtained a minimum thickness of 159 nm.

To ensure the photoresist thickness remained below 160 nm before exposure and thus minimized the influence of thickness on laser projection patterns, we spun the resist at 8000 rpm for 60 seconds. We have now included the spin-coating curve (Fig. R12) of the measured thickness for AR-N-7520 in the manuscript. Given that this thickness is well below the exposure wavelength.

Fig. R12 Thickness of AR-N-7520 at different spin-coating speeds.

[1] Jeon, T.Y., Park, S.-G., Kim, D.-H. and Kim, S.-H., Standing-Wave-Assisted Creation of Nanopillar Arrays with Vertically Integrated Nanogaps for SERS-Active Substrates. *Adv. Funct. Mater.*, **25**: 4681-4688(2015).

Comment 3:

The method has only been demonstrated for 2D lithography; this limits its applicability range.

Thus, applications highlighted by the authors, including microelectronic integrated circuits and optical waveguides, predominantly focus on 2D structures. In this regard, the method offers large-surface applicability in 2D multiphoton lithography, enabling the use of readily available low-power lasers.

The methodology is demonstrated using a single commercially available monomer, AR-N-7520, chosen for its absorption properties and directly excited and cross-linked without any modifications (no additional description provided by the authors, I assume there are no photoinitiators inside). Exploring the method's applicability to more complex photoresists would be valuable, as most 3D printing photoresists consist of a resin and photoinitiator that absorbs light (followed by for e.g. a radical formation) and triggers cross-linking reactions. In many cases, photoresists have lower viscosity, resulting in significant diffusivity of photoinitiators, radicals, and other reactive species; however, upon irradiation, the polymerization process initiates, rapidly reducing the diffusivity (by orders of magnitude) of these excited or radical-forming species (DOI: 10.1364/OE.19.019486, 10.1117/12.2287470). Reducing irradiation power, as it has been introduced in fpTPA, slows down the crosslinking process, allowing photoinitiators and radicals to diffuse more quickly, potentially increasing feature size, as observed for e.g. in STED-inspired lithography concepts (10.1002/lpor.201100046 DOI: 10.1117/12.2287470). The major and strong limitation of this method is that it has not yet been demonstrated with other photoresists, thus restricting its general applicability to monomers that can be directly excited and applied as low-viscosity monolayers. Thus, I strongly recommend incorporating experimental data and additional paragraphs that demonstrates/address the impact of diffusion when using other low-viscosity photoresists to generalize the applicability of this method in 2D and 3D applications.

Response:

Thank you for your detailed feedback and for highlighting the application scope. In our study, we focused on validating the few-photon absorption (fpTPA) process using a non-chemically amplified, negative-tone photoresist in 2D planar structures, as it demonstrates the effectiveness of the fpTPA approach. This optical projection method can also be applied to 3D printing; however, it faces additional challenges, such as limitations imposed by the properties of the photoresist material and resolution degradation caused by dispersion along the z-axis. At higher precision requirements for 3D fabrication, the absorption mechanisms of fpTPA may improve the accuracy of 3D processing. In future work, we will continue to explore methods to address the challenges associated with 3D structure

fabrication. We acknowledge the importance of exploring different photoresists in nanolithography, especially considering the diffusion of photoinitiators and radicals within photoresist layers. As noted in Yaoyu Cao's study [1], photoinitiators generate radicals upon photon absorption, which initiate monomer polymerization. In irradiated areas, a photo-inhibition effect can hinder radical diffusion and polymerization. However, at the edges of these regions, where light intensity is lower, the inhibition weakens, allowing radicals to diffuse and trigger polymerization.

The AR-N-7520 photoresist we used does not contain photoinitiators, nor does it require a post-exposure bake. They employ non-chain growth mechanisms to initiate the desired chemical transformation when exposed to light [2] As described in section S9 Photopolymerization, the commercial photoresist used here is a mixture of a bisazide cross-linker and Novolak resin rather than a monomer. As the n-CARs do not require any additional chemical amplification, they are devoid of the most serious problem that almost all CARs face i.e. acid diffusion in the solid state causing considerable line-edge roughness (LER) and line-width roughness (LWR) [3]. The polymerization mechanism, as shown in our prior work, involves the elimination of nitrogen from bisazides under femtosecond laser irradiation, forming a reactive nitrene [4]. This nitrene then inserts into the phenolic resin with assistance from an alkyl group (Fig. R13).

Fig. R13 AR-N-7520 photoresist polymerization mechanism

As you recommended, we have included experimental results involving additional photoresists. Given the low-power characteristics of *fp*TPA, traditional photoresists with photoinitiators could affect clarity and introduce complex diffusion and polymerization effects. Specifically, we fabricated various microstructures using not only AR-N 7520, but also other materials such as SU-8, AR4340 (chemical amplified negative resist), AR5350 (non-chemical amplified positive resist), SCR500 (liquid radical resist), and Silver/Polymer nanocomposites. The diffusion introduced by the chemically amplified resist effectively enhanced the efficiency of TPDOP, for example, the acid diffusion of AR5350 in post baking process led to one order lower the irradiated photon density comparing with that of the n-CA resist, AR-N 7520. These experiments provide a broader perspective on the versatility of this method. Moreover, we have added a corresponding discussion on Page 9 (Line 14) of the revised manuscript: “We have fabricated various microstructures using diverse photoresists besides ARN 7520 to generalize the applicability of this method, such

as SU-8, AR-N-4340, AR-N-5350, SCR500 and Silver/Polymer nanocomposite (see Supplementary S19 and Figure S17 for details). These results demonstrate the wide application possibilities for potential application.”

[1] Cao Y, Gan Z, Jia B, Evans RA, Gu M. High-photosensitive resin for super-resolution direct-laser-writing based on photoinhibited polymerization. *Opt. Express*, Sep 26;19(20):19486-94 (2011).

[2] S. Gauci, A. Vranic, E. Blasco, S. Bräse, M. Wegener, C. Barner - Kowollik, Photochemically Activated 3D Printing Inks: Current Status, Challenges, and Opportunities, *Adv. Mater.*, **36** (2023).

[3] S. Ghosh, P. Parameswaran, S. Sharma, P. Reddy, S. Pal, K. Gonsalves, Recent advances in non-chemically amplified photoresists for next generation IC technology, *RSC Adv.*, **6** (2016)

[4] Yu-Huan Liu, Yuan-Yuan Zhao, Feng Jin, Xian-Zi Dong, Mei-Ling Zheng, Zhen-Sheng Zhao, and Xuan-Ming Duan. *Nano Lett.*, **21**(9), 3915-3921 (2021).

[5] Rong-Rong Wang, Mei-Ling Zheng, Wei-Cai Zhang, Jie Liu, Teng Li, Xian-Zi Dong, and Feng Jin, *Nano Lett.*, **22** (24), 9823-9830, (2022)

Comment 4:

What is the DMD-frequency? How fast can one switch between projection-patterns?

Response:

The DMD frequency refers to the rate at which the DMD (Digital Micromirror Device) can load projection patterns per second. Each mirror on the DMD can flip within 5 microseconds. Combined with our home-made control program, we can configure the frame rate for loading images, and set specific dark and light times for individual images. This enables a tunable projection frequency ranging from 1 Hz to 22.7 kHz.

Comment 5:

Suppl. Mats: the absorption spectrum of the AR-N-7520 is shown to be between 250 – 500nm. Given that two-photon excitation occurs at 400 nm, I would expect the spectrum to begin at 200 nm. Please include both excitation wavelengths in the spectrum diagram. How big is the 1-photon absorbance of the AR-N-7520 @ 400 and @517nm? Since it does not look a pure 2-photon absorption, I suggest changing the title of the manuscript to multiphoton. I think an additional information on the possible impact of the 1-photon absorption for the used monomer shall be discussed (see for e.g. DOI: 10.1002/adom.202000895). I believe it will have minimal impact on fpTPA, but since the authors compare fpTPA excitation to "high-power" polymerization, addressing this might create additional value to the manuscript.

Response:

Thank you for your thoughtful suggestion. In response to your comment, we have re-tested the one-photon absorption spectrum of AR-N-7520 and updated it in the supplementary materials (see Figure R14 (Figure S8)). The commercial photoresist primarily consists of Bisazide cross-linker and Novolak resin. From the absorption spectra provided ($Abs_{(400)} = 0.0411$, $Abs_{(517)} = 0.0067$), we observe that the one-photon absorption

at both 400 nm and 517 nm is very low. The very low absorbance around 400 nm and 517 nm could belong to the absorption of Novolak resin, since the pure Novolak resin generally also has weak absorption in visible region.

Thank you for your thoughtful suggestion regarding the manuscript title. Generally Multiphoton Absorption is used to describe higher order absorption. After careful consideration, we have decided to maintain the original title, as it specifically reflects the focus of our study on two-photon absorption (TPA) and the unique approach we have presented using low-power laser irradiation for nanoprinting. While we acknowledge that the process may involve multiple-photon absorption, we believe that emphasizing "two-photon absorption" in the title more accurately reflects the primary mechanism being utilized in this work.

Figure R14 Updated AR-N-7520 absorption spectrum (190-800 nm, with sampling test conducted on Shimadzu UV-3600i Plus).

[1] Fu Li et al. ,3D printing of inorganic nanomaterials by photochemically bonding colloidal nanocrystals. *Science*, **381**,1468-1474(2023).

Minor:

Comment 1:

Supplementary information p.4: please provide a graphical description for virtual state, the ground state, the intermediate state, and the transition state.

Response:

The description in the SI file has been updated to consistent with the main manuscript, specifically aligning with the data presented in Figure 1B. The state of the data now matches the corresponding information provided in the manuscript. Accordingly, we have revised the description in S3 of Supplementary information.

Comment 2:

fJ/pulse pixel – add brackets

Response:

Thank you for your valuable suggestion. We have revised the expression and replaced the unit with fJ/(pulse·pixel) for better clarity.

Comment 3:

manuscript @ 332 : Finally, it is noteworthy that the distribution of TPA induced by few-photon irradiation has been narrowed down to the nanometer scale, independent of the light wavelength. – I believe the authors mean that the results are independent of the two wavelengths used.

Response:

Thank you for your comment. We have revised the sentence for clarity on **Page9 Line32** in the revised manuscript:

"Finally, it is noteworthy that the distribution of TPA induced by few-photon irradiation has been narrowed down to the nanometer scale, independent of the two specific wavelengths used."

Comment 4:

Manuscript @ 172: ... under few-photon irradiation can significantly can significantly ...

Response:

Thank you for your comment. We have corrected it in the revised manuscript.

Comment 5:

Fig 1. Please add definitions of the letters as in Fig. 2.

Response:

Thank you for your suggestion. We have added the definitions of the letters in Figure 1.

Comment 6:

Please rephrase the statement; it is a bit confusing or add a sketch describing the molecular states since it is not full consistent with Fig.1 (regarding the 3 states- bring in connection to the virtual state) : ' There are three states for an active molecule in space, namely the ground state, the intermediate state, and the transition state. The area can no longer absorb the photon in the transition state. The active molecule in the ground state will transit after absorbing one photon. To reach the intermediate state, the active molecule absorbs another photon and jumps to the transition state."

Response:

Thank you for your careful reading and for pointing out the inconsistency. We have revised the text in the manuscript to ensure consistency with the SI. The statement on **S3** in the revised supplementary is as follows:

"There are three states for an active molecule in space, namely the initial state, the intermediate state, and the excited state. The area can no longer absorb the photon in the

excited state. The active molecule in the initial state will transit after absorbing one photon. To reach the intermediate state, the active molecule absorbs another photon and jumps to the excited state." We believe this revision clarifies the explanation.

Reviewer #3 (Remarks to the Author):**Comment 1:**

Two-photon absorption under few-photon irradiation for optical nanoprinting

This work explores the concept of few-photon irradiated two-photon absorption (fpTPA) and its application for optical nanoprinting. Overall, it shows nice fabrication results, but the field of TPA/lithography is moving rapidly, and so it is important to put these results in perspective.

Response to Comment 1:

Thank you for your comment. We have added a discussion on recent advancements in two-photon absorption (TPA) lithography in manuscript, along with a comparative analysis between femtosecond TPA (fsTPA) lithography and other high-efficiency TPA methods. To demonstrate the advantages of our fabrication results, we calculated the throughput values representing "efficiency" and provided a detailed comparison with similar methods and their relevant parameters (as shown in Supporting Information S18). The table S4 shows that our proposed method achieves high levels in both linewidth resolution and line spacing resolution, and its efficiency is on the same order of magnitude as similar methods, approximately 10^6 voxels/s. In our research, we focus on 2D patterning of photoresists, where "throughput" is defined as the area exposed per minute, with units in mm^2/min . We have calculated the throughput values in our experimental process and updated them in the manuscript (see Line 31 Page 7) and supplementary (S10). The method for calculating throughput is equal to the exposed field area per exposure / effective exposure time per field.

Comment 2:

The concept of fpTPA is introduced as novel and potentially distinct from "traditional" two-photon absorption (TPA), but to my understanding fpTPA is just simply TPA occurring under lower photon fluxes. In practice, all practitioners in the field reduce the pulse energy so that the centre of the spatial intensity profile is just above the nonlinear polymerisation threshold. This, of course, allows for the smallest possible voxels to be created, and hence allows high-resolution 3D structuring. However, the material response is generally more important than the photon spatial and temporal distribution, as each photoresist has its own response (and hence achievable 3D resolution). I didn't see the material response discussed in detail. For example, I didn't see any mention of the chemical chain length or quenching properties. Perhaps my understanding here is limited, but I was not able to understand the novelty of fpTPA vs the standard TPA.

Response to Comment 2:

Thank you for your comment. In conventional or standard two-photon absorption (TPA) lithography, researchers often reduce pulse energy or exposure dose so that the center of the spatial light intensity distribution just exceeds the threshold, aiming to achieve the

smallest “voxel” [1]. However, this approach has limitations when dealing with free-radical polymerization photoresists, as radical diffusion and oxygen quenching effects tend to “blur” the smallest voxel units, making it challenging to achieve minimal voxel sizes [2,3]. Our proposed *fs*TPA lithography differs from standard TPA by precisely controlling the probability of single or few-photon TPA interactions with the material within a single pulse. We employ a non-chemically amplified photoresist based on a stepwise polymerization mechanism (each TPA event initiates one cross-linking reaction) [4]. Since there is no significant acid diffusion or quenching effect, the location of each cross-linking reaction is exact, determined by the spatial probability distribution of TPA photons. Consequently, an extremely weak photon flux can confine the cross-linking reaction to a minimal region at the center of the light intensity distribution. We add the details of the materials and its response in polymerization process described on page 5 (Line 30) of the manuscript: “We chose a commercially available non-chemically amplified (non-CA) negative photoresist (AR-N 7520) because its degree of polymerization, driven by the stepwise photopolymerization mechanism, is easily controllable and quantifiable for N_{eTPA} under few-photon irradiation.” Specifically, we fabricated various microstructures using not only AR-N 7520, but also other materials such as SU-8, AR4340(chemical amplified negative resist), AR5350 (non-chemical amplified positive resist), SCR500 (liquid radical resist), and Silver/Polymer nanocomposites. The diffusion introduced by the chemically amplified resist effectively enhanced the efficiency of TPDOPL, for example, the acid diffusion of AR5350 in post baking process led to one order lower the irradiated photon density comparing with that of the n-CA resist, AR-N 7520. These experiments provide a broader perspective on the versatility of this method. Moreover, we have added a corresponding discussion on Page 9 (Line 14) of the revised manuscript.

Reference:

1. J. Fischer, M. Wegener, Three - dimensional optical laser lithography beyond the diffraction limit, *Laser Photonics Rev.*, **7**, 22-44 (2013).
2. A. Alubaidy, K. Venkatakrisnan, B. Tan, Dual wavelength multiphoton absorption, *Des. Monomers Polym.*, **17**, 126-131 (2014).
3. A. Pikulin, N. Bityurin, Spatial resolution in polymerization of sample features at nanoscale, *Physical Review B*, **75**, 195430 (2007).
4. Y. H. Liu, Y. Y. Zhao, F. Jin, et al., $\lambda/12$ super resolution achieved in maskless optical projection nanolithography for efficient cross-scale patterning, *Nano Lett.*, **21**, 3915-3921 (2021).

Comment 3:

The authors propose a spatiotemporal model to describe TPA under few-photon irradiation. I’m not an expert in quantum, but this idea of a “real” virtual state conflicts with my understanding of the processes involved. It would be interesting to see evidence for this “real” virtual state existing in practice.

Response to Comment 3:

Thank you for your comment. The “virtual state” is an intermediate state to describe the two-photon absorption process. The process of two-photon absorption, first described

theoretically in 1931 by Maria Göppert-Mayer [1], involves the simultaneous absorption of two photons (of energies $h\nu_1$ shown in Fig. 1 of manuscript) by a material system, for example a molecule, that occurs when the energy $h\nu_1 + h\nu_1$ is in resonance with one of the electronic states of the system. This process can be thought of as an initial interaction of a photon (the first one) of energy $h\nu_1$ with the molecule, which is thus left in a temporary virtual state of energy $h\nu_1$ above the ground state S_0 (similar to what occurs in Rayleigh scattering or nonresonant Raman scattering) [2,3]. This is not a real state (eigenstate) of the molecule and it exists only for a short time interval, τ . If during τ another one photon (the second one) of energy $h\nu_1$ interacts with the molecule, it can be excited to state S_1 . In quantum mechanics research, this “virtual state” is considered to be an unmeasurable state and has nothing to do with the measurement accuracy of the instrument [4]. The order of magnitude for τ , which can be estimated from the uncertainty principle, is 10^{-15} – 10^{-16} s for photon energies in the visible and near-IR ranges [5,6]. The qualifier “simultaneous” for TPA is used to indicate that the two photons interact with the molecule within the time τ and that no real states act as an intermediate state in this process.

Reference:

- [1] M. Göppert - Mayer, Elementary processes with two quantum transitions, *Annalen der Physik*, 18, 466-479 (2009) (Göttinger Dissertation. Reprint of “Über Elementarakte mit zwei Quantensprüngen,” *Annalen der Physik*, 9, 273 – 294, 1931).
- [2] W. L. Peticolas, Multiphoton spectroscopy, *Annu. Rev. Phys. Chem.*, **18**, 233-260 (1967).
- [3] W. M. McClain, Two-photon molecular spectroscopy, *Acc. Chem. Res.*, **7**, 129-135 (1974).
- [4] Masters BR. "Historical Development of Non-linear Optical Microscopy and Spectroscopy". In Masters BR, So P (eds.). *Handbook of Biomedical Nonlinear Optical Microscopy*[M]. US: Oxford University Press (2008).
- [5] Xu C, Webb W W. Multiphoton excitation of molecular fluorophores and nonlinear laser microscopy[M]//*Topics in Fluorescence Spectroscopy: Volume 5: Nonlinear and Two-Photon-Induced Fluorescence*. Boston, MA: Springer US, 471-540 (2002).
- [6] F. Boitier, A. Godard, E. Rosencher, et al., Measuring photon bunching at ultrashort timescale by two-photon absorption in semiconductors, *Nat. Phys.*, **5**, 267-270 (2009).

Comment 4:

The resolution of the structures is not analysed correctly, as this requires Fourier analysis to identify resolution. See for example, Figure 3c) and e), where the “thinnest” point of the line is used. It is not a fair result to just take the thinnest line width.

Response to Comment 4:

Thanks for the comment. This would help us improve the quality of our manuscript. We have adopted a more rigorous Fourier analysis method to evaluate the actual resolution of fpTPA lithography.

We can more accurately evaluate the spatial frequency characteristics of the lithographic structure [1-3] using Fourier analysis, thereby obtaining a true resolution indicator. As you pointed out, relying solely on the minimum line width is not enough, as this may ignore the impact of edge effects and other factors on resolution. Through Fourier

analysis, the quality of the lithographic structure can be evaluated more comprehensively, including edge roughness, line width deviation and other indicators, thereby obtaining a more reliable resolution evaluation.

We have calculated the roughness parameters in the figure and added further discussion in the Supplementary Information (S12). The added description of the line width in the revised text “The smallest line width is 26 nm and the average line width is 43 nm with standard deviation of 7 nm and roughness of 3-4 nm.”

Your suggestion is very valuable, and we will pay more attention to using rigorous methods such as Fourier analysis to evaluate the resolution performance of lithography in future analyses. This will help us to better understand the limitations and development potential of this technology.

[1] Ohfujii T, Endo M, Morimoto H. Theoretical analysis of line-edge roughness using FFT techniques[C]//Advances in Resist Technology and Processing XVI. SPIE, **3678**: 732-738 (1999).

[2] A. Yamaguchi, O. Komuro, Characterization of line edge roughness in resist patterns by using Fourier analysis and auto-correlation function, *Jpn. J. Appl. Phys.*, **42**, 3763 (2003).

[3] Lawrence W. Spatial frequency analysis of line-edge roughness in nine chemically related photoresists[C]// The International Society for Optical Engineering, SPIE, **5039**: 713-724 (2003).

Comment 5:

The concept of using multiple lines to “get around” the diffraction limit is quite an established technique, including specifically using a DMD with multiple patterns for two-photon lithography, for achieving “sub diffraction limit” resolution. It would be good to put the novelty of your approach into perspective regarding the existing demonstrations of this approach in the literature.

Response to Comment 5:

Thank you for your comments. In traditional lithography, engineers typically employ multiple patterning (MP) techniques to push beyond lithographic resolution limits. This approach involves multiple development and etching steps, which inevitably introduce alignment errors. In contrast, multiple exposure (ME) generally requires only a single development step, making the process simpler and more cost-effective. However, ME is ineffective in conventional UV lithography, as single-photon exposure doses are linearly additive during multiple exposures, meaning that repeated exposures cannot enhance resolution [1]. On the other hand, there are several concepts for special double-exposure techniques that employ optical material nonlinearity to achieve sub-diffraction-limit imaging for dense line-spaces [2].

In this manuscript, we combine ME with the two-photon absorption (TPA) effect. During multiple exposures, the exposure dose in TPA is nonlinearly additive. At the central point between two adjacent light fields, which would just be resolvable with single-photon exposure, the intensity is $I(0) + I(0) = 1$. However, with two-photon exposure, the central

point intensity becomes $I^2(0) + I^2(0) = 0.5$, achieving complete resolution. Thus, implementing a multiple exposure strategy in TPA lithography can significantly decrease line pitch and achieve “sub-diffraction limit” resolution. We experimentally demonstrate, for the first time, this material-driven nonlinear effect as a resolution enhancement technique.

To highlight the novelty of our approach, we have added further discussion and citations (page 7 Line 14): “It’s a dual-exposure technique that eliminates the need for alignment to enhance lithographic resolution in TPDOPL. Using computer-controlled DMD to generate low spatial frequency, sparse 'digital mask' patterns and alternating dual exposures, we double the density of nanopatterns. Since the spacing between DMD micromirrors is fixed, alignment errors are eliminated, making multiple exposures on a single photoresist coating feasible without physical mask alignment steps — surpassing the diffraction limit achievable with single exposures in traditional lithography.”

Reference:

[1] Andreas Erdmann. Optical and EUV Lithography: A Modeling Perspective[M]. Publisher. SPIE; Publication date. March 2(2021).

[2] S. Lee, K. Jen, C. G. Willson, et al., Materials modeling and development for use in double-exposure lithography applications, *J. Micro/Nanolithogr. MEMS MOEMS*, **8** (2009).

Comment 6:

“This result indicates that the resolution of TPA under few-photon irradiation can significantly surpass the diffraction limit of the employed wavelength.” Again, this isn’t a demonstration of surpassing the diffraction limit, in the precise definition. Diffraction limit refers to the resolvability of two point sources, not the smallest size of a feature, or the separation of two features when more than one pattern is used.

Response to Comment 6:

Thank you for your comments. We fully agree with the reviewer’s clarification on the definition of the “diffraction limit.” Indeed, the discussion in our manuscript regarding the analysis of surpassing the diffraction limit was somewhat simplified. To avoid ambiguity, we have revised the descriptions in the manuscript (see page 7 line 8) and added related discussions in supplementary (S13).

We also need to add further explanation regarding lithographic resolution. Lithographic resolution is generally expressed as $R/CD=k_1\lambda/NA$, which has two implications. First, strictly speaking, $R=k_1\lambda/NA$ represents the line-spacing resolution (as the reviewer noted as “the resolvability of two point sources”), equivalent to half the smallest resolvable pitch of two adjacent line patterns (half-pitch). Second, in traditional lithography processes, the photoresist pattern is typically required to have a width-to-gap ratio of 1:1, where the width corresponds to the critical dimension (CD). To achieve this 1:1 width-to-gap ratio in resolvable pattern fabrication, the exposed linewidth CD must be controlled to be pitch/2, which gives $CD=width=gap=HP=k_1\lambda/NA$, thus expressing the linewidth resolution. Therefore, the lithographic resolution $R/CD=k_1\lambda/NA$ can describe both the linewidth limit and the pitch limit.

Lithography resolution:

$$R/CD = k_1 \frac{\lambda}{NA} \left. \begin{array}{l} \sim R \text{ (HP, half pitch) } k_1 \geq 0.25 \\ \sim CD \text{ (width : gap=1:1)} \end{array} \right\}$$

Width: the minimum width of line
Pitch: the distance of adjacent lines

Fig. R2 Definition and diagram of photolithography resolution (w: Width; g: Gap; p: Pitch)

In the early studies on two-photon lithography, the focus was primarily on narrowing the width of polymer lines, and thus researchers often evaluated the advancement of lithography results based on achieving widths smaller than the diffraction limit. Later, the development of STED-two-photon lithography further reduced polymer line widths, leading some researchers to discuss the limit of line spacing in TPA lithography. In our manuscript, we used a laser with a center wavelength of 517 nm and an objective lens with an NA of 1.49. On one hand, the minimum width of the exposed structure was 26 nm, equivalent to $(\sim 0.075) \lambda/NA$, which is indeed smaller than the diffraction limit. On the other hand, the minimum line spacing was 210 nm = $(\sim 0.605) \lambda/NA$, also smaller than the diffraction limit.

Comment 7:

In general, the results are less impressive in resolution than others in the literature, particularly given that the results here are in 2D, and not the expected 3D that other's are demonstrating. It would be good to clarify this in more detail.

Response to Comment 7:

Thank you for your comments. The traditional serial two-photon lithography that has achieved critical dimensions (CD) as small as sub-10 nm [1]. However, it is essential to emphasize that our proposed fsTPA lithography is a high-efficiency parallel lithography technique, and thus cannot be directly compared to conventional two-photon lithography (including STED two-photon lithography) [2]. Among currently reported parallel lithography techniques, our study achieves one of the best trade-offs between resolution and lithography area (see Table S4 in SI). At an average lithography efficiency ($\sim 10^6$ voxels/s), to the best of our knowledge, the resolution values reported in our manuscript are the highest in terms of both linewidth and pitch resolution in two-photon projection lithography. Notably, only reference [3] in the field of parallel lithography has discussed the limits of pitch resolution.

Furthermore, the fsTPA lithography mechanism we proposed is designed to achieve extreme lithographic CD. Thinner resist films generally enhance the yield of high-resolution structures. For example, in the field of electron beam lithography, the CD achievable is typically inversely correlated with resist thickness, such that thinner films allow for smaller CDs [4]. Likewise, our fsTPA lithography technique could also be extended to 3D lithography, which would require improvements to resist formulations and exposure systems. In the future, we plan to explore research related to 3D lithography applications.

Reference:

[1] S. Wang, Y. Yu, H. Liu, et al., Sub-10-nm suspended nano-web formation by direct laser

writing, *Nano Futures*, **2**, 025006 (2018).

[2] J. Fischer, M. Wegener, Three-dimensional optical laser lithography beyond the diffraction limit, *Laser Photonics Rev.*, **7**, 22-44 (2013).

[3] S. K. Saha, D. Wang, V. H. Nguyen, et al., Scalable submicrometer additive manufacturing, *Science*, **366**, 105-109 (2019).

[4] A. E. Grigorescu, C. W. Hagen, Resists for sub-20-nm electron beam lithography with a focus on HSQ: state of the art, *Nanotechnol.*, **20**, 292001 (2009).

Point-to-point Response to Reviewers

Reviewer #1 (Remarks to the Author):

The authors have responded to all the points raised by the reviewers, making changes and providing strong rebuttals. I find their arguments convincing and believe that the article is suitable for publication after addressing the following minor points.

- The image quality in Figure S17m is very poor. Hard to read text.
- Scalebars in Figure S17b are illegible.
- Table S4 has different units for fabrication rate. Please make them consistent. The authors should be able to determine an area/s rate for the volume printing methods.

Response:

Thanks to the reviewer's affirmation and recognition, we have updated Suppl. Fig.17 and unified the expression of fabrication rate in Suppl. Table4.

Reviewer #2 (Remarks to the Author):

In the revised manuscript titled "Two-photon absorption under few-photon irradiation for optical nanoprinting," improved significantly, however it requires some minor revisions:

1.) I suggest adding paper citations:

(<http://dx.doi.org/10.1364/OE.24.027077>, DOI: 10.1039/D0NA00154F) - In this context, the crosslinking degree of the monomers is controlled by the exposure dose.

Response:

Thanks to the reviewer's affirmation and recognition, we have updated fig suppl 17 and unified the expression of fabrication rate in Table S4.

Reviewer #3 (Remarks to the Author):

I am happy that the authors have addressed my concerns, and my recommendation is to accept the manuscript.

Response:

Thank the reviewer for their recognition!